# Φ-DVAE: Learning Physically Interpretable Representations with Nonlinear Filtering

## Abstract

Incorporating unstructured data into physical models is a challenging problem that is emerging in data assimilation. Traditional approaches focus on well-defined observation operators whose functional forms are typically assumed to be known. This prevents these methods from achieving a consistent model-data synthesis in configurations where the mapping from data-space to model-space is unknown. To address these shortcomings, in this paper we develop a physics-informed dynamical variational autoencoder (Φ-DVAE) for embedding diverse data streams into time-evolving physical systems described by differential equations. Our approach combines a standard (possibly nonlinear) filter for the latent state-space model and a VAE, to embed the unstructured data stream into the latent dynamical system. A variational Bayesian framework is used for the joint estimation of the embedding, latent states, and unknown system parameters. To demonstrate the method, we look at three examples: video datasets generated by the advection and Korteweg-de Vries partial differential equations, and a velocity field generated by the Lorenz-63 system. Comparisons with relevant baselines show that the Φ-DVAE provides a data efficient dynamics encoding methodology that is competitive with standard approaches, with the added benefit of incorporating a physically interpretable latent space.

## 1 Introduction

Physical models — as represented by ordinary, stochastic, or partial differential equations — are ubiquitous throughout engineering and the physical sciences. These differential equations are the synthesis of scientific knowledge into mathematical form. However, as a description of reality they are imperfect (Judd & Smith, 2004), leading to the well-known problem of model misspecification (Box, 1979). At least since Kalman (1960) physical modellersls with observations (Anderson & Moore, 1979). Such approaches are usually either solving the inverse problem of attempting to recover model parameters from data, and/or, the data assimilation (DA) problem of conducting state inference based on a time-evolving process.

For the inverse problem, Bayesian methods are common (Tarantola, 2005; Stuart, 2010). In this, prior belief in model parameters $\mathbf{\Lambda}$ is updated with data $\mathbf{y}$ to give a posterior distribution, $p(\mathbf{\Lambda}|\mathbf{y})$. This describes uncertainty with parameters given the data and modelling assumptions. DA can also proceed from a Bayesian viewpoint, where inference is cast as a nonlinear state-space model (SSM) (Law et al., 2015; Reich & Cotter, 2015). The SSM is typically the combination of a time-discretised differential equation and an observation process: uncertainty enters the model through extrusive, additive errors. For a latent state variable $\mathbf{u}_n$ representing some discretised system at time $n$, with observations $\mathbf{y}_n$, the object of interest is the filtering distribution $p(\mathbf{u}_n|\mathbf{y}_{1:n})$, where $\mathbf{y}_{1:n} := \{\mathbf{y}_k\}_{k=1}^n$. Additionally, the joint filtering and estimation problem, which estimates $p(\mathbf{u}_n, \mathbf{\Lambda}|\mathbf{y}_{1:n})$ has received significant attention in the literature (see, e.g., Kantas et al. (2015) and references therein). This has been well studied in, e.g., electrical engineering (Storvik, 2002), geophysics (Bocquet & Sakov, 2013), neuroscience (Ditlevsen & Samson, 2014), chemical engineering (Kravaris et al., 2013), biochemistry (Dochain, 2003), and hydrology (Moradkhani et al., 2005), to name a few.

Typically in data assimilation tasks, while parameters of an observation model may be unknown, the observation model itself is assumed known (Kantas et al., 2015). This assumption breaks down in settings where data arrives in various modalities, such as videos, images, or audio, hindering the

ability to perform inference. However, in such cases often the underlying variation in the data stream is due to a latent physical process, which is typically at least partially known.

In this work, these data streams are video data and velocity fields. We develop a variational Bayes (VB) (Blei et al., 2017) methodology which jointly solves the inverse and filtering problems for the case in which the observation operator is unknown. We model this unknown mapping with a variational autoencoder (VAE) (Kingma & Welling, 2014), which encodes the assumed time-dependent observations $\mathbf{y}_{1:N}$ into pseudo-data $\mathbf{x}_{1:N}$ in a latent space. On this latent space, we stipulate that the pseudo-observations are taken from a known dynamical system, given by a stochastic ordinary differential equation (ODE) or partial differential equation (PDE) with possibly unknown coefficients. The differential equation is also assumed to have stochastic forcing, which accounts for possible model misspecification. The stipulated system gives a structured prior $p(\mathbf{x}_{1:N}|\mathbf{\Lambda})$, which acts as a physics-informed regulariser whilst also enabling inference over the unknown $\mathbf{\Lambda}$. This prior is approximated using classical nonlinear filtering algorithms. Our framework is fully probabilistic: inference proceeds from a derived evidence lower bound (ELBO), enabling joint estimation of unknown network parameters and unknown dynamical coefficients via VB. To set the scene for this work, we now review the relevant literature.

## 2 RELATED WORK

As introduced above, VAEs (Kingma & Welling, 2014) are a popular high-dimensional encoder. A VAE defines a generative model that learns low-dimensional representations, $\mathbf{x}$, of high-dimensional data, $\mathbf{y}$, using VB. To perform efficient inference, a variational approximation $q_\phi(\mathbf{x}|\mathbf{y})$ is made to the intractable posterior $p(\mathbf{x}|\mathbf{y})$. Variational parameters $\phi$ are estimated via optimisation of the ELBO. This unsupervised learning approach infers latent representations of high-dimensional data. Recent works have extended the VAE to high-dimensional time-series data $\mathbf{y}_{1:N}$, indexed by time $n$, with the aim of jointly learning latent representations $\mathbf{x}_{1:N}$, and a dynamical system that evolves them. These dynamical variational autoencoder (DVAE) methods (Girin et al., 2021) enforce the dynamics with a structured prior $p(\mathbf{x}_{1:N})$ on the latent space.

Various DVAE methods have been proposed. The Kalman variational autoencoder (KVAE) of Fraccaro et al. (2017) is a popular approach, which encodes $\mathbf{y}_{1:N}$ into latent variables $\mathbf{x}_{1:N}$ that are assumed to be observations of a linear Gaussian state-space model (LGSSM), driven by latent dynamic states $\mathbf{u}_{1:N}$. Assumed linear dynamics are jointly learnt with the encoder and decoder, via Kalman filtering/smoothing. Another approach is the Gaussian process variational autoencoder (GPVAE) (Pearce, 2020; Jazbec et al., 2021; Fortuin et al., 2020), which models $\mathbf{x}_{1:N}$ as a temporally correlated Gaussian process (GP). The Markovian variant of Zhu et al. (2022) allows for a similar Kalman procedure as in the KVAE, except, in this instance, the dynamics are known and are given by an stochastic differential equation (SDE) approximation to the GP (Hartikainen & Sarkka, 2010). A related approach is provided for control applications in Watter et al. (2015); Hafner et al. (2019), where locally linear embeddings are estimated. Yildiz et al. (2019) also propose the so-called ODE$^2$VAE, which encodes the data to an initial condition which is integrated through time using a Bayesian neural ODE (Chen et al., 2018). This trajectory, only, is used to generate the reconstructions via the decoder network.

A related class of methods are deep SSMs (Bayer & Osendorfer, 2014; Krishnan et al., 2015; Karl et al., 2017). These works assume that the parametric form of the SSM is unknown, and replace the transition and emission distributions with neural network (NN) models, which are trained based on an ELBO. They harness the representational power of deep NNs to directly model transitions between high-dimensional states. More emphasis is placed on generative modelling and prediction than representation learning, or system identification. We also note the related VAE works of Wu et al. (2021); Franceschi et al. (2020); Babaeizadeh et al. (2022), which use VAE-type architectures for similar video prediction tasks. In Chung et al. (2015) the variational recurrent neural networks (VRNN) attempt to capture variation in highly structured time-series data, by pairing a recurrent NN for learning nonlinear state-transitions with a sequential latent random variable model.

Methods to include physical information inside of autoencoders have been studied in the physics community. A popular approach uses SINDy (Brunton et al., 2016) for discovery of low-dimensional latent dynamical systems using autoencoders (Champion et al., 2019). A predictive framework is given in Lopez & Atzberger (2021), which aims to learn nonlinear dynamics by jointly optimizing

an ELBO. Following our notation, this learns some function which maps $\mathbf{u}_n \mapsto \mathbf{u}_{n+k}$, for some $k$, via a VAE. Lusch et al. (2018) use a physics-informed autoencoder to linearise nonlinear dynamical systems via a Koopman approach; inference is regularised through incorporating the Koopman structure in the loss function. Otto & Rowley (2019) present a similar method, and an extension of these approaches to PDE systems is given in Gin et al. (2021). Morton et al. (2018) use the linear regression methods of Takeishi et al. (2017) within a standard autoencoder to similarly compute the Koopman observables. Erichson et al. (2019) derive an autoencoder which incorporates a linear Markovian prediction operator, similar to a Koopman operator, which uses physics-informed regulariser to promote Lyapunov stability. Hernández et al. (2018) studies VAE methods to encode high-dimensional dynamical systems. Finally, we note the related work which studies the estimation of dynamical parameters within so-called "gray-box" systems, blending NN methods with known physical laws (Lu et al., 2020; Yin et al., 2021; Long et al., 2018; de Bézenac et al., 2019).

**Our contribution.** In this paper we propose a physics-informed dynamical variational autoencoder ($\Phi$-DVAE): a DVAE approach which imposes the additional structure of known physics on the latent space. We assume that there is a low-dimensional dynamical system generating the high-dimensional observed time-series: a NN is used to learn the unknown embedding to this lower dimensional space. On the lower-dimensional space, the embedded data are pseudo-observations of a latent dynamical system, which is, in general, derived from a numerical discretisation of a nonlinear PDE. However, the methodology is suitably generic, allowing for ODE latent systems. Inference on this latent system is done with efficient nonlinear stochastic filtering methods, enabling the use of mature DA algorithms within our framework. Our approach follows a probabilistically coherent VB construction and allows for joint learning of both the embedding and unknown dynamical parameters.

Instead of learnt, linear dynamics with the KVAE (Fraccaro et al., 2017), $\Phi$-DVAE assumes a possibly misspecified nonlinear differential equation is driving the variation in the latent space, with possibly unknown parameters. This is in contrast to incorporating generic physical structure in the latent space (such as Koopman structure (Lusch et al., 2018), or Lyapunov stability (Erichson et al., 2019)), or generic temporal structure (such as in the GPVAE (Pearce, 2020; Jazbec et al., 2021; Fortuin et al., 2020)). Our latent dynamical systems give a known functional form of the latent transition density, instead of the learnt, NN-parameterised, transition and emission densities in deep SSMs (Bayer & Osendorfer, 2014; Krishnan et al., 2015; Karl et al., 2017). Similarly, whilst we share commonality with a latent differential equation, the $\Phi$-DVAE differs with the ODE$^2$VAE (Yildiz et al., 2019) as we perform inference with this ODE/PDE, instead of learning it and leveraging it to deterministically evolve the latents. Our approach trades off the generality of latent system discovery against the ability to infer physical quantities of interest relating to a particular latent system. $\Phi$-DVAE can infer physical parameters and states, solving the joint filtering and parameter estimation problem in scenarios where the observation model is unknown.

## 3 THE PROBABILISTIC MODEL

In this section we define our probabilistic model; our presentation roughly follows the structure of the generative model. We first give an overview of the dependencies between variables, as described by conditional probabilities. We then cover the latent differential equation model used to describe the underlying physics. Then, the pseudo-observation model is covered, followed by the decoder and the encoder. To be precise, we assume a general SSM:

$$\mathbf{\Lambda} \sim p(\mathbf{\Lambda}), \ \mathbf{u}_n | \mathbf{u}_{n-1}, \mathbf{\Lambda} \sim p(\mathbf{u}_n | \mathbf{u}_{n-1}, \mathbf{\Lambda}), \ \mathbf{x}_n | \mathbf{u}_n \sim p_\nu(\mathbf{x}_n | \mathbf{u}_n), \ \mathbf{y}_n | \mathbf{x}_n \sim p_\theta(\mathbf{y}_n | \mathbf{x}_n),$$

where $\mathbf{\Lambda}$ describes the parameters of the Markov process $\{\mathbf{u}_n\}_{n=0}^N$ evolving w.r.t. the *dynamic model* $p(\mathbf{u}_n | \mathbf{u}_{n-1}, \mathbf{\Lambda})$, $\nu$ describes the parameters of the *likelihood* denoted by $p_\nu(\mathbf{x}_n | \mathbf{u}_n)$, and $\theta$ describes NN parameters for the *decoder* $p_\theta(\mathbf{y}_n | \mathbf{x}_n)$. The conditional independence structure imposed by the model gives

$$p(\mathbf{y}_{1:N}, \mathbf{x}_{1:N}, \mathbf{u}_{1:N}, \mathbf{\Lambda}) = p_\theta(\mathbf{y}_{1:N} | \mathbf{x}_{1:N}) p_\nu(\mathbf{x}_{1:N} | \mathbf{u}_{1:N}) p(\mathbf{u}_{1:N} | \mathbf{\Lambda}) p(\mathbf{\Lambda}). \tag{1}$$

Intuitively, $\{\mathbf{y}_n\}_{n=1}^N$ is the sequence of high-dimensional video frames, $\{\mathbf{x}_n\}_{n=1}^N$ is its embedding (or the pseudo-data), and $\{\mathbf{u}_n\}_{n=0}^N$ is the *latent physics process*. For each $n$, we assume that $\mathbf{y}_n \in \mathcal{Y}$ (with $\dim(\mathcal{Y}) = n_y$), $\mathbf{x}_n \in \mathbb{R}^{n_x}$, $\mathbf{u}_n \in \mathbb{R}^{n_u}$, and $\mathbf{\Lambda} \in \mathbb{R}^{n_\lambda}$. In what follows, we describe the components of our probabilistic model in detail.

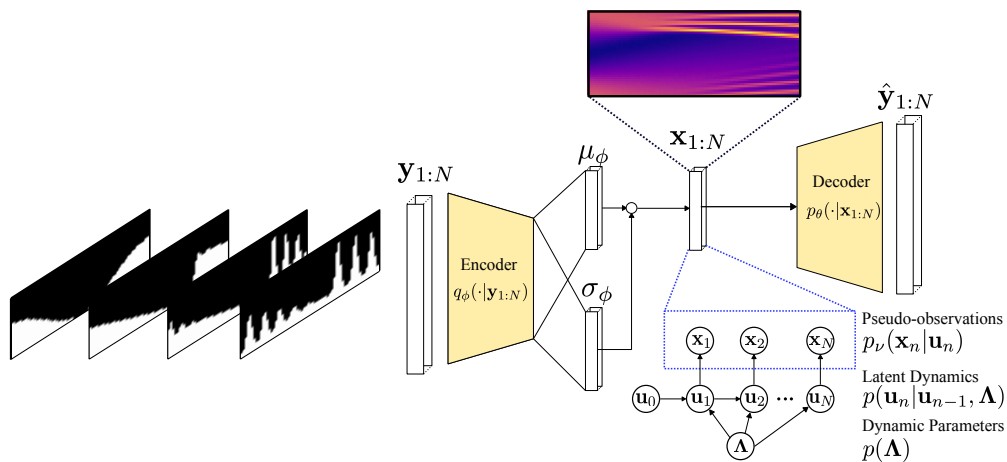

Figure 1: An illustration of the Φ-DVAE model. On the left, the frames of a video can be seen which are denoted $\mathbf{y}_{1:N}$. These are converted into physically interpretable low-dimensional encodings $\mathbf{x}_{1:N}$ using an encoder. This learning process is informed by the physics driven state-space model which treats $\mathbf{x}_{1:N}$ as *pseudo-observations*, which can be seen on the bottom right.

## 3.1 DYNAMIC MODEL

The first component of the generative model is the latent dynamical system $p(\mathbf{u}_n|\mathbf{u}_{n-1}, \mathbf{\Lambda})$. In general, we model this latent physics process $\{\mathbf{u}_n\}_{n=0}^N$ as a discretised stochastic PDE, however ODE latent physics is admissible within our framework (see Section 5.1). We discretise this process with the statistical finite element method (STATFEM) (Girolami et al., 2021; Duffin et al., 2021; 2022; Akyildiz et al., 2022), forming the basis of the physics-informed prior on the latent space. Stochastic additive forcing inside the PDE represents additive model error, which results from possibly misspecified physics. Full details, including the ODE case, are given in Appendix C.

In general, we assume that the model has possibly unknown coefficients $\mathbf{\Lambda}$; on these we place the Bayesian prior $\mathbf{\Lambda} \sim p(\mathbf{\Lambda})$ (Stuart, 2010), describing our *a priori* knowledge on the model parameters before observing any data. We also assume that $\mathbf{u}_0$, the initial condition, is known up to measurement noise, with prior $p(\mathbf{u}_0)$ set accordingly. For pedagogical purposes, we derive the discrete-time dynamic model using the Korteweg-de Vries (KDV) equation as a running example, which is used in later sections as an example PDE:

$$\partial_t u + \alpha u \partial_s u + \beta \partial_s^3 u = \dot{\xi}, \quad \dot{\xi} \sim \mathcal{GP}(0, \delta(t - t') \cdot k(s, s')), \tag{2}$$

where $u := u(s,t) \in \mathbb{R}$, $\xi := \xi(s,t)$, $s \in [0, L]$, $t \in [0, T]$, and $\mathbf{\Lambda} = \{\alpha, \beta\}$. Informally $\dot{\xi}$ is a GP, with delta correlations in time, and spatial correlations given by the covariance kernel $k(\cdot, \cdot) \colon \mathbb{R} \times \mathbb{R} \to \mathbb{R}$ (Williams & Rasmussen, 2006). This is an uncertain term in the PDE, representing possible model misspecification. Note that we assume all GP hyperparameters are known in this work. The KDV equation is used to model nonlinear internal waves (see, e.g., Drazin & Johnson, 1989), and describes the balance between nonlinear advection and dispersion. Note that although the KDV equation defines a scalar field, $u(s,t)$, the approach similarly holds for vector fields.

Following STATFEM, the equations are first spatially discretised with the finite element method (FEM), then discretised in time. Thus, equation 2 is multiplied with a smooth test function $v(s) \in V$, where $V$ is an appropriate function space, and integrated over the domain $\Omega$ to give the weak form (Brenner & Scott, 2008; Thomée, 2006)

$$\langle \partial_t u, v \rangle + \alpha \langle u \partial_s u, v \rangle + \beta \langle \partial_s^3 u, v \rangle = \langle \dot{\xi}, v \rangle,$$

where $\langle \cdot, \cdot \rangle$ is the $L^2(\Omega)$ inner product.

The domain is discretised to give the mesh $\Omega_h$ with vertices $\{s_j\}_{j=0}^{n_h}$. In this case, we take the $s_j$ to be a uniformly spaced set of points, so that $s_j = jh$ ($h$ gives the spacing between grid-points). On the mesh a set of polynomial basis functions $\{\phi_j(s)\}_{i=1}^{n_u}$ is defined, such that approximation to the

PDE. Letting $u_h(s,t) = \sum_{i=1}^{n_u} u_i(t)\phi_i(s)$, the weak form is now rewritten with these basis functions

$$\langle \partial_t u_h, \phi_j \rangle + \alpha \langle u \partial_s u_h, \phi_j \rangle + \beta \langle \partial_s^3 u_h, \phi_j \rangle = \langle \dot{\xi}, \phi_j \rangle, \quad j = 1, \ldots, n_u.$$

This gives a finite-dimensional SDE over the FEM coefficients $\mathbf{u}(t) = (u_1(t), \ldots, u_{n_u}(t))^\top$:

$$\mathbf{M}\frac{\mathrm{d}\mathbf{u}}{\mathrm{d}t} + \beta\mathbf{A}\mathbf{u} + \alpha\mathcal{F}(\mathbf{u}) = \dot{\boldsymbol{\xi}}, \quad \dot{\boldsymbol{\xi}}(t) \sim \mathcal{N}(\mathbf{0}, \delta(t - t') \cdot \mathbf{G}),$$

where $\mathbf{M}_{ij} = \langle \phi_i, \phi_j \rangle$, $\mathbf{A}_{ji} = \langle \partial_s^3 \phi_i, \phi_j \rangle$, $\mathcal{F}(\mathbf{u})_j = \langle u_h \partial_s u_h, \phi_j \rangle$, and $\mathbf{G}_{ij} = \langle \phi_i, \langle k(\cdot, \cdot), \phi_j \rangle \rangle$. Time discretisation eventually gives the transition density $p(\mathbf{u}_n | \mathbf{u}_{n-1}, \Lambda)$, for $\mathbf{u}_n = \mathbf{u}(n\Delta_t)$, whose form is dependent on the discretisation used (see Appendix C for all details).

## 3.2 Likelihood

The second component of the generative model is the *likelihood* $p_\nu(\mathbf{x}_n | \mathbf{u}_n)$. This density acts as a *data likelihood* for pseudo-data $\{\mathbf{x}_n\}_{n=1}^N$. This middle layer in the model is usually necessary, as high-dimensional observations $\{\mathbf{y}_n\}_{n=1}^N$ may only be generated by some observed dimensions of $\{\mathbf{u}_n\}_{n=0}^N$. For example, perhaps it is known *a priori* that only a single component of a latent coupled differential equation generates the observations (see also Section 5.1). This explicit likelihood is introduced to separate the encoding process from the state-space inference; in practice we compute the pseudo-data via the encoding $q_\phi(\mathbf{x}_{1:N} | \mathbf{y}_{1:N})$, then condition on it with standard nonlinear filtering algorithms (Fraccaro et al., 2017).

The latent states $\mathbf{u}_n$ are mapped at discrete times to pseudo-observations via $\mathbf{x}_n = \mathbf{H}\mathbf{u}_n + \mathbf{r}_n$, with $\mathbf{r}_n \sim \mathcal{N}(\mathbf{0}, \mathbf{R})$. We parameterise this observation density as $p_\nu(\mathbf{x}_n | \mathbf{u}_n)$ where $\nu = \{\mathbf{H}, \mathbf{R}\}$. Both the pseudo-observation operator $\mathbf{H} \in \mathbb{R}^{n_x \times n_u}$ and the noise covariance $\mathbf{R} \in \mathbb{R}^{n_x \times n_x}$ are assumed to be known in this work. An additional noise process is assumed, $\mathbf{r}_n$, to represent extraneous uncertainty associated with the pseudo-observations. Observations $\mathbf{y}_n$ are related to pseudo-observations $\mathbf{x}_n$ via the decoder, represented with the conditional density $p_\theta(\mathbf{y}_n | \mathbf{x}_n)$. The combination of the transition and observation densities provides the nonlinear Gaussian SSM:

$$\text{Transition:} \quad \mathbf{u}_n = \mathcal{M}(\mathbf{u}_{n-1}) + \mathbf{e}_{n-1}, \quad \mathbf{e}_{n-1} \sim \mathcal{N}(\mathbf{0}, \mathbf{Q}),$$
$$\text{Pseudo-observation:} \quad \mathbf{x}_n = \mathbf{H}\mathbf{u}_n + \mathbf{r}_n, \quad \mathbf{r}_n \sim \mathcal{N}(\mathbf{0}, \mathbf{R}).$$

Inference is performed with the extended Kalman filter (ExKF) (Jazwinski, 1970; Law et al., 2015), which computes $p(\mathbf{u}_n | \mathbf{x}_{1:n}, \Lambda)$. Further details are contained in Section 4.

## 3.3 Decoder

The last component of our generative model is the decoder $p_\theta(\mathbf{y}_n | \mathbf{x}_n)$, which describes the unknown mapping between the pseudo-observations $\mathbf{x}_n$, and the observed data $\mathbf{y}_n$. The decoding of latents to data should model as closely as possible the true data generation process. Prior knowledge about this process can be used to select an appropriate $p_\theta(\mathbf{y}_n | \mathbf{x}_n)$. No temporal structure is assumed on $\theta$, so the decoder is shared across all times $p_\theta(\mathbf{y}_{1:N} | \mathbf{x}_{1:N}) = \prod_{n=1}^N p_\theta(\mathbf{y}_n | \mathbf{x}_n)$. For more details on specific architectures see Appendix A.

## 4 Variational Inference

In this section, we introduce the variational family and the ELBO. Denote by $q(\mathbf{u}_{1:N}, \mathbf{x}_{1:N}, \Lambda | \mathbf{y}_{1:N})$ the variational posterior, which, similar to Fraccaro et al. (2017), factorises as

$$q(\mathbf{u}_{1:N}, \mathbf{x}_{1:N}, \Lambda | \mathbf{y}_{1:N}) = q(\mathbf{u}_{1:N} | \mathbf{x}_{1:N}, \Lambda) q_\phi(\mathbf{x}_{1:N} | \mathbf{y}_{1:N}) q_\lambda(\Lambda).$$

Note here that we do not make a variational approximation $q(\mathbf{u}_{1:N}, \mathbf{x}_{1:N}, \Lambda | \mathbf{y}_{1:N})$; this is taken to be the exact posterior $p(\mathbf{u}_{1:N} | \mathbf{x}_{1:N}, \Lambda)$. We derive the ELBO to be (see also Appendix B)

$$\log p(\mathbf{y}_{1:N}) \geq \int \log\left[\frac{p(\mathbf{u}_{1:N}, \mathbf{x}_{1:N}, \Lambda, \mathbf{y}_{1:N})}{q(\mathbf{u}_{1:N}, \mathbf{x}_{1:N}, \Lambda | \mathbf{y}_{1:N})}\right] q(\mathbf{u}_{1:N}, \mathbf{x}_{1:N}, \Lambda | \mathbf{y}_{1:N}) \mathrm{d}\mathbf{x}_{1:N} \mathrm{d}\mathbf{u}_{1:N} \mathrm{d}\Lambda$$

$$= \mathbb{E}_{q_\phi}\left[\log \frac{p_\theta(\mathbf{y}_{1:N} | \mathbf{x}_{1:N})}{q_\phi(\mathbf{x}_{1:N} | \mathbf{y}_{1:N})} + \mathbb{E}_{q_\lambda}\left[\log p(\mathbf{x}_{1:N} | \Lambda) + \log \frac{p(\Lambda)}{q_\lambda(\Lambda)}\right]\right]. \tag{3}$$

Typically this expectation is not analytically tractable and Monte Carlo is used to compute an approximation.

**Nonlinear Filtering.** In the ELBO of equation 3, estimating $\log p(\mathbf{x}_{1:N}|\mathbf{\Lambda})$ requires marginalising over $\mathbf{u}_{1:N}$, *the latent physics process*. We perform this via the EXKF (Jazwinski, 1970; Law et al., 2015), which recursively computes a Gaussian approximation to the filtering posterior $p(\mathbf{u}_n|\mathbf{x}_{1:n},\mathbf{\Lambda}) \approx \mathcal{N}(\mathbf{m}_n,\mathbf{C}_n)$. We note however that this can also be realised with other nonlinear filters, e.g., ensemble Kalman filters (Chen et al., 2022) or particle filters (Corenflos et al., 2021). The factorisation of the pseudo-observation marginal likelihood, $p(\mathbf{x}_{1:N}|\mathbf{\Lambda}) = p(\mathbf{x}_1|\mathbf{\Lambda})\prod_{n=2}^{N} p(\mathbf{x}_n|\mathbf{x}_{1:n-1},\mathbf{\Lambda})$, enables computation, as the filter can compute $\log p(\mathbf{x}_n|\mathbf{x}_{1:n-1},\mathbf{\Lambda})$, at each prediction step, via $p(\mathbf{x}_n|\mathbf{x}_{1:n-1},\mathbf{\Lambda}) = \int p(\mathbf{x}_n|\mathbf{u}_n,\mathbf{\Lambda})p(\mathbf{u}_n|\mathbf{x}_{1:n-1},\mathbf{\Lambda})\,\mathrm{d}\mathbf{u}_n$.

**Variational approximations.** As with the decoder, encoder parameters $\phi$ are shared between variational distributions $\{q_\phi(\mathbf{x}_n|\mathbf{y}_n)\}_{n=1}^{N}$ to give an amortized approach (Kingma et al., 2019). Unless otherwise specified, for each $n$ the encoding has the form $q_\phi(\mathbf{x}_n|\mathbf{y}_n) = \mathcal{N}(\mu_\phi(\mathbf{y}_n), \sigma_\phi(\mathbf{y}_n))$. Functions $\mu_\phi(\mathbf{y}_n)\colon \mathbb{R}^{n_y} \to \mathbb{R}^{n_x}$ and $\sigma_\phi(\mathbf{y}_n)\colon \mathbb{R}^{n_y} \to \mathbb{R}^{n_x}$ are NNs, with parameters $\phi$ to be learnt. Specific encoding architectures are given in Appendix A. As for $q_\lambda$, we set it to a Gaussian with mean $\mu_\lambda$ and variance $\mathrm{diag}(\sigma_\lambda)$.

Latent Dynamics: $\mathbf{u}_{1:N}$

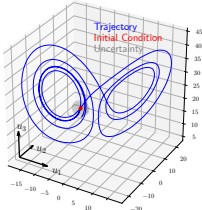

Pseudo-Observations: $\mathbf{x}_{1:N}$

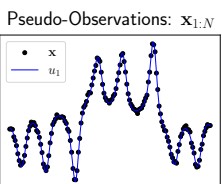

Velocity Field: $\mathbf{y}_N$

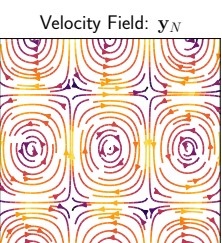

Figure 2: Lorenz-63: latent states $\mathbf{u}_{1:N}$, pseudo-observations $\mathbf{x}_{1:N}$, and velocity field $\mathbf{y}_N$.

# 5 EXPERIMENTS

We present three examples with different dynamical systems. To demonstrate the generality of the method, the first uses the stochastic Lorenz-63 system (Lorenz, 1963), a highly nonlinear stochastic ODE. In this case, high-dimensional observations are of a velocity field being modulated by the chaotic system. For the final two examples, we use video data. We consider the advection and KDV PDEs, and, in these examples, we indirectly observe high-dimensional representations in the form of video data. This mimics the experimental setup of the various DVAE papers (e.g., Fraccaro et al., 2017; Pearce et al., 2018; Jazbec et al., 2021; Fortuin et al., 2020; Zhu et al., 2022; Girin et al., 2021). For these two examples, video datasets are generated in a similar fashion. In both cases, simulated data emulate the scenario where a noisy video of an internal wave profile is captured. Internal waves arise as waves of depression or elevation flowing within a density-stratified fluid at regions of maximum density gradient (Gerkema & Zimmerman, 2008). Our experiment setup thus emulates an idealised setup where a black-and-white, side-on, video of a laboratory experiment has been obtained, and is inspired by scenarios where the high-resolution use of classical measurement devices is not feasible, yet the use of commonplace video-capturing devices is (see, e.g., Horn et al., 2001; 2002).

The advection equation example is motivated by an internal wave propagating undisturbed through some medium. For this linear case, comparisons with the KVAE reveal that after training, the Φ-DVAE outperforms both in terms of the estimated ELBO and in terms of the mean-squared-error (MSE). The KDV example is a more complex case, and extends into the nonlinear PDE setting, while also being a classical model for internal waves (Drazin & Johnson, 1989). For this example, comparisons are made with VRNNs, the GPVAE, and the standard VAE. We demonstrate that the MSE of the Φ-DVAE is comparable or better than these approaches. Furthermore, for joint state and parameter inference, we verify the methodology and demonstrate contraction of the posterior about the truth.

## 5.1 LORENZ-63 EXAMPLE

In our first example, the latent dynamical model $p(\mathbf{u}_n|\mathbf{u}_{n-1},\mathbf{\Lambda})$ is given by an Euler-Maruyama discretisation (Kloeden & Platen, 1992) of the stochastic Lorenz-63 system,

$$
\begin{aligned}
\mathrm{d}u_1 &= (-\sigma u_1 + \sigma u_2)\mathrm{d}t + \mathrm{d}w_1, \\
\mathrm{d}u_2 &= (-u_1 u_3 + r u_1 - u_2)\mathrm{d}t + \mathrm{d}w_2, \\
\mathrm{d}u_3 &= (u_1 u_2 - b u_3)\mathrm{d}t + \mathrm{d}w_3,
\end{aligned}
\tag{4}
$$

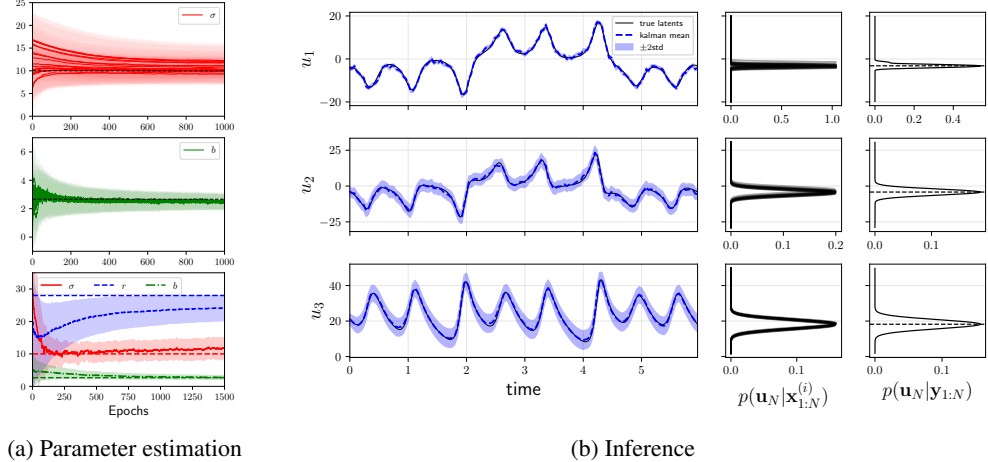

(a) Parameter estimation          (b) Inference

Figure 3: Lorenz-63 system: (a) the results of parameter estimation for (top) $\sigma$ estimation, (center) $b$ estimation, and (bottom) joint estimation of $\mathbf{\Lambda} = \{\sigma, r, b\}$. In (b) we show an example result of state estimation. Left column is the true Lorenz states vs. ExKF means, center and right columns show the distribution of the estimate of the final states.

where $\mathbf{u}(t) := [u_1(t), u_2(t), u_3(t)]^\top, t \in [0, 6]$, $\mathbf{u}_n = [u_1(n\Delta_t), u_2(n\Delta_t), u_3(n\Delta_t)]$, $\mathbf{\Lambda} = \{\sigma, r, b\}$, and $w_1$, $w_2$, and $w_3$ are independent Brownian motion processes (Øksendal, 2003; Särkkä & Solin, 2019). For full details we refer to Appendices A and C. The Lorenz-63 system is classical system widely used to benchmark filtering and data assimilation methods (see, e.g., Akyildiz & Míguez, 2020). It was popularised in Lorenz (1963) through its characterisation of "deterministic nonperiodic flow", and is a common example of chaotic dynamics.

We observe synthetic 2D velocity fields, $\mathbf{y}_{1:N}$, of convective fluid flow, and we use our method to embed these synthetic data into the stochastic Lorenz-63 system. The Lorenz-63 system is related to the velocity fields through a truncated spectral expansion (see, e.g., Wouters, 2013). In brief, it is assumed that the velocity fields have no vertical velocity, so the 3D velocity field is realised in 2D. The velocity field can be described by the stream function $\psi := \psi(s_1, s_2, t)$, where $s_1$ and $s_2$ are the spatial coordinates of variation, respectively, and thus $\mathbf{y}(t) = (-\partial_{s_2}\psi, 0, \partial_{s_1}\psi)$. A truncated spectral approximation and a transform to non-dimensional equations yields $\psi(s_1, s_2, t) \propto u_1(t) \sin(\pi s_1/l) \sin(\pi s_2/d)$, where $u_1(t)$ is governed by the Lorenz-63 ODE (i.e., equation 4 with $w_i \equiv 0$).

To generate the synthetic data $\mathbf{y}_{1:N}$, we generate a trajectory $\mathbf{u}_{1:N}^{\text{true}}$ from the deterministic version of 4, at discrete timepoints $n\Delta_t$, and use the generated $u_{1,n}^{\text{true}}$ to compute the two-dimensional velocity field, $\mathbf{y}_n$, via $\psi(s_1, s_2, t)$. This corresponds to having a middle layer $x_n = u_{1,n} + w_n$ where $w_n \sim \mathcal{N}(0, R^2)$, with likelihood $p(x_n|\mathbf{u}_n) = \mathcal{N}(\mathbf{h}^\top \mathbf{u}_n, R^2)$ where $\mathbf{h} = [1, 0, 0]^\top$. The synthetic data of $\mathbf{u}^{\text{true}}(t)$, $\mathbf{x}_{1:N}$, and $\mathbf{y}_{1:N}$ are visualised in Figure 2; the full trajectory $\mathbf{u}^{\text{true}}(t)$ is shown in 3D, and the velocity field $\mathbf{y}_n$ is shown in 2D, for a single $n$. The decoding is assumed to be of the form $p_{\mathbf{w}}(\mathbf{y}_n|x_n) = \mathcal{N}(\mathbf{w}x_n, \eta^2\mathbf{I})$, with unknown coefficients $\mathbf{w} \in \mathbb{R}^{n_y}$. The variational encoding $q_{\mathbf{w}}(x_n|\mathbf{y}_n)$ is determined via a pseudo-inverse, as detailed in Appendix A, along with the relevant hyperparameters and numerical details.

Figure 3a displays parameter estimation results. For individual $\sigma$ and $b$ estimation (Figure 3a, top and center), we initialise the variational posterior randomly, and visualise each across training epochs. Both results show the posteriors contracting about the true value, with $b$ demonstrating more rapid convergence, visually. For joint estimation (Figure 3a, bottom), similar behaviour is observed, with the parameter $r$ not identified by the final epoch. We conjecture this is due to identifiability with other parameters when estimating jointly. Note, however, that the true values are all contained within the confidence bands of the final variational posteriors.

We also investigate the posterior inference achieved by $\Phi$-DVAE. We visualise the filtering posterior through time, conditioned on a sample from the trained encoder $\mathbf{x}_{1:N}^{(i)} \sim q_\phi(\cdot|\mathbf{y}_{1:N})$, with fixed,

known $\mathbf{\Lambda}$. Figure 3b (left) shows clear agreement between the filter mean and the latent states $\mathbf{u}_{1:N}^{\text{true}}$. Marginalising over the encoding (Figure 3b (centre)) targets the filtering posterior directly conditioned on observed data $\mathbf{y}_{1:N}$, and demonstrates unbiased mean estimates of the true state. This is particularly clear for the first latent dimension, where the posterior conditioned on an individual sample $\mathbf{x}_{1:N}^{(i)}$ often has poor posterior coverage of the true value (cf. Figure 3b (right)).

## 5.2 Advection PDE

As a second example, we consider the advection equation with periodic boundary conditions. In this example, we derive the transition density $p(\mathbf{u}_n|\mathbf{u}_{n-1}, \mathbf{\Lambda})$ from a STATFEM discretisation of a stochastic advection equation:

$$\partial_t u + c\,\partial_s u = \dot{\xi}, \quad \dot{\xi} \sim \mathcal{GP}(0, \delta(t - t') \cdot k(s, s')), \tag{5}$$

where $u := u(s,t)$, $\xi := \xi(s,t)$, $s \in [0,1]$, $t \in [0,40]$, and $u(s,t) = u(s+1,t)$. Recall that, as in Section 5.2, the FEM coefficients $\mathbf{u}_n = (u_1(n\Delta_t), \ldots, u_{n_u}(n\Delta_t))$ are the latent variables. These are related to the discretised solution via $u_h(s, n\Delta_t) = \sum_{i=1}^{n_u} u_i(n\Delta_t)\phi_i(s)$ (see Appendix A and C).

Video data $\mathbf{y}_{1:N}$ is generated from the deterministic version of equation 5 (i.e. equation 5 with $\xi \equiv 0$). A trajectory $\mathbf{u}_{1:N}^{\text{true}}$ is simulated and the corresponding FEM solutions $u_h^{\text{true}}(s, n\Delta_t)$ are imposed onto a 2D grid. On the grid, pixels below $u_h^{\text{true}}(s, n\Delta_t)$ are lit-up in a binary fashion, with salt-and-peper noise (Gonzalez & Woods, 2007). In this experiment we use fixed parameters, setting $\mathbf{\Lambda} \equiv c = 0.5$. We set the decoder to $p_\theta(\mathbf{y}_n|\mathbf{x}_n) = \text{Bern}(\mu_\theta(\mathbf{x}_n))$ and the encoder as $q_\phi(\mathbf{x}_n|\mathbf{y}_n) = \mathcal{N}(\mu_\phi(\mathbf{y}_n), \sigma_\phi(\mathbf{y}_n))$. As previous, see Appendix A for full details.

Due to linearity of the underlying dynamical system, we compare the $\Phi$-DVAE to the KVAE for a set of video data generated from the advection equation, for various dimensions of the KVAE latent space. Specifying a particular form of latent dynamics on the latent states increases the inductive bias imposed on the latent space, and should provide faster learning — and more likely representations — than with learnt dynamics.

To explore this, we compare our method to KVAE for the linear advection example in terms of the ELBO and normalised MSE, over training epochs (all methods use Adam (Kingma & Ba, 2017)). These are plotted in Figure 4. For the MSE, KVAE quickly learns to reconstruct the images, with $\Phi$-DVAE taking longer to reconstruct with similar accuracy but eventually producing better reconstructions (final MSEs 0.0221 vs. 0.0533 for the $\Phi$-DVAE, KVAE-64, respectively). The ELBO for $\Phi$-DVAE is rapidly optimised in comparison to the KVAE models, and is greater by the end of training. The trained $\Phi$-DVAE gives better evidence for the data (greater ELBO), whilst providing more accurate reconstructions (lower MSE).

## 5.3 Korteweg–de Vries PDE

Our final example uses the KDV equation as the underlying dynamical system. As previously, the latent transition density $p(\mathbf{u}_n|\mathbf{u}_{n-1}, \mathbf{\Lambda})$ defines the evolution of the FEM coefficients, as given by a STATFEM discretisation of a stochastic KDV equation:

$$\partial_t u + \alpha u \partial_s u + \beta \partial_s^3 u = \dot{\xi}, \quad \dot{\xi} \sim \mathcal{GP}(0, \delta(t - t') \cdot k(s, s')),$$

where $u := u(s,t)$, $\xi := \xi(s,t)$, $s \in [0,2]$, $t \in [0,1]$, and $u(s,t) = u(s+2,t)$. Parameters are $\mathbf{\Lambda} = \{\alpha, \beta\}$.

Data is simulated in the same fashion as in the advection equation: we simulate a trajectory $\mathbf{u}_{1:N}^{\text{true}}$ using a FEM discretisation of the deterministic KDV equation and we impose FEM solutions $u_h^{\text{true}}(s, n\Delta_t)$

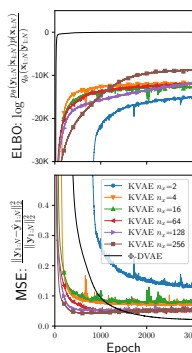

Figure 4: Advection: ELBO (*top*) and reconstruction MSE (*bottom*), against ground truth, for ($\Phi$-DVAE) against KVAE.

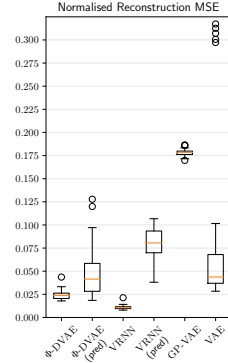

Figure 5: KDV: normalised reconstruction MSE after 200 epochs, for 30 independent simulations.

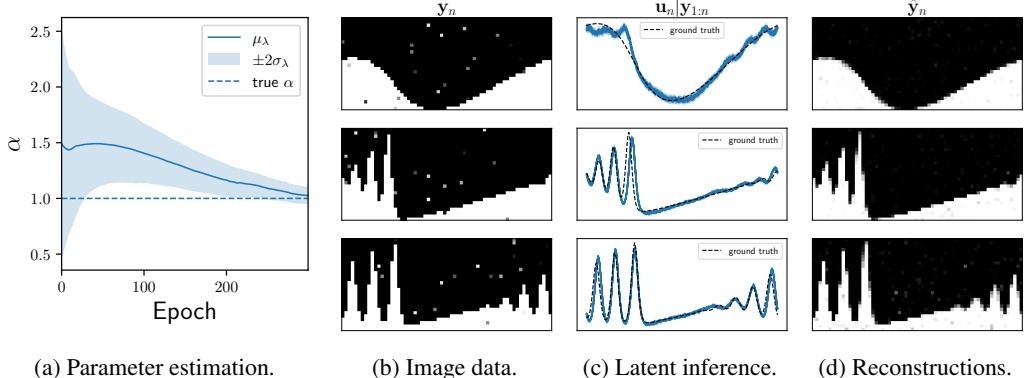

(a) Parameter estimation.  (b) Image data.  (c) Latent inference.  (d) Reconstructions.

Figure 6: KDV joint inference results; frames shown for $n = \{10, 50, 75\}$.

on a 2D grid. We light up pixels below the solution, and add salt-and-pepper noise. For the data-generating process, we use the parameters $\boldsymbol{\Lambda} = \{\alpha = 1, \beta = 0.022^2\}$, and the initial condition of $u_h^{\text{true}}(s, 0) = \cos(\pi s)$ as in the classical work of Zabusky & Kruskal (1965). This regime is characterised by the steepening of the initial condition and the generation of solitons; nonlinear waves which have particle-like interactions (Drazin & Johnson, 1989). As for the advection example, we set the decoder as $p_\theta(\mathbf{y}_n | \mathbf{x}_n) = \text{Bern}(\mu_\theta(\mathbf{x}_n))$ and the encoder as $q_\phi(\mathbf{x}_n | \mathbf{y}_n) = \mathcal{N}(\mu_\phi(\mathbf{y}_n), \sigma_\phi(\mathbf{y}_n))$; again, see Appendix A for details.

We test the VRNN, GPVAE and the standard VAE approaches alongside $\Phi$-DVAE with equal encoding dimension $n_x = 40$. In Figure 5 we report the normalised MSE after a fixed number of epochs, using Adam (Kingma & Ba, 2017) with the same learning rate for each method. $\Phi$-DVAE outperforms both GPVAE and the standard VAE in terms of median MSE. Note the variation, in MSE, of the standard VAE is also greater than the other models, suggesting that the dynamical structure provides more consistent learning. Both $\Phi$-DVAE and VRNN perform well, producing visually similar reconstructions, with the VRNN having lower median MSE (0.0239 vs 0.0105, respectively). A predictive MSE is also computed by giving the model a single image frame to encode, sampling from the generative model forward in time, and comparing the decoded samples against the ground truth images. Here, $\Phi$-DVAE outperforms VRNN with median predictive MSEs 0.0415 vs 0.0806 respectively.

We report joint estimation results for the partially known KDV PDE, where we fix $\beta = 0.022^2$, and estimate $\alpha$. The prior over $\alpha$, $p(\alpha) = \mathcal{N}(1.5, 0.3^2)$, is semi-informative. Joint inference of $\alpha$ and latent states $\mathbf{u}_n$ is shown in Figure 6. The Gaussian variational posterior, $q_\lambda(\alpha) = \mathcal{N}(\mu_\lambda, \sigma_\lambda^2)$ (initialised at the prior), contracts about the true value $\alpha = 1.0$ (see Figure 6a). The filtering posterior, $p(\mathbf{u}_n | \mathbf{y}_{1:n})$, is shown in Figure 6c. This is obtained via Monte Carlo approximation, marginalising over the variational posteriors $q_\phi, q_\lambda$, to account for uncertainty in the encoding and parameter estimates. Including a structured prior on the latent space has forced the encoding to be representative of observations taken from the KDV system, clearly capturing the latent dynamics causing the variation in the image data. Figures 6b and 6d display the image data and reconstructions respectively, showing the ability of $\Phi$-DVAE to both accurately reconstruct and de-noise the data.

## 6 CONCLUSION

In this paper we developed $\Phi$-DVAE, a methodology that allows for the incorporation of unstructured data into physical models, in settings where the model-data mapping may be unknown. The proposed approach utilizes variational autoencoders and nonlinear filtering algorithms for PDEs, to learn physically interpretable latent spaces where analysis and prediction can be performed straightforwardly. Our framework connects traditional nonlinear filtering techniques and VAEs, opening up the possibility of further combinations of these methods. Future work will focus on more challenging PDE systems, as well as more complex, and higher-dimensional, observational data.

## REPRODUCIBILITY STATEMENT

We have included full numerical details to reproduce all results in our paper in Appendix A. The code we have developed for this paper will be made publicly available after the decision, enabling the generation of all datasets and figures.

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

## A  NUMERICAL DETAILS AND NETWORK ARCHITECTURES

**Lorenz-63 experiment.**

- Time-series length: $N = 150$.
- Input: $n_y = 200$.
- Pseudo-observations: $n_x = 1$.
- Latent $n_u = 3$.
- Decoder: $p_{\mathbf{w}}(\mathbf{y}_n | x_n) = \mathcal{N}(\mathbf{w}x_n, \eta^2 \mathbf{I})$, $\eta = 0.005$.
- Encoder: $q_{\mathbf{w}}(x_n | \mathbf{y}_n) = \mathcal{N}((\mathbf{w}^\top \mathbf{w})^{-1} \mathbf{w}^\top \mathbf{y}_n, \eta^2 (\mathbf{w}^\top \mathbf{w})^{-1})$
- Latent initial condition: $\mathbf{u}_0 = [-3.7277, -3.8239, 21.1507]^\top$
- Latent noise processes: $\mathbf{L} = \mathrm{diag}(0.2^2)$, $\mathbf{R} = \mathrm{diag}(0.4^2)$.
- Latent discretisation: Euler-Maruyama, $dt = 0.001$.
- Joint parameter prior: $p([\sigma, r, b]) = \mathcal{N}([30, 20, 5]^\top, \mathrm{diag}([12^2, 10^2, 3^2]^\top))$.
- Optimiser: Adam, learning rate $= 10^{-4}$.

To generate the data, we simulate the Lorenz SDE with $dt = 0.001$ and take pseudo-observations $x_n$ every 40 time-steps of the latent system, for a total of $N = 150$ with $\Delta_t = 0.04$. Velocity measurements are taken in the $s_1$ and $s_2$ directions over a regular $10 \times 10$ grid on the domain $s_1, s_2 \in [-4, 4]$, via the streamfunction $\psi(s_1, s_2, t)$. These measurements are flattened to the data vector $\mathbf{y}_n \in \mathbb{R}^{n_y}$, $n_y = 200$. Parameters for data generation are $\mathbf{\Lambda} = \{\sigma = 10, r = 28, b = 8/3\}$.

**Advection equation experiment.**

- Time-series length: $N = 200$.
- Input: $n_y = 784$.
- Pseudo-observations: $n_x = 64$.
- Latent $n_u = 64$.
- Decoder: $p_\theta(\mathbf{y}_n | \mathbf{x}_n) = \mathrm{Bern}(\mu_\theta(\mathbf{x}_n))$
- $\mu_\theta(\cdot)$: MLP, two fully connected hidden layers with dimension 128
- Encoder: $q_\phi(\mathbf{x}_n | \mathbf{y}_n) = \mathcal{N}(\mu_\phi(\mathbf{y}_n), \sigma_\phi(\mathbf{y}_n))$
- $\mu_\phi(\cdot), \sigma_\phi(\cdot)$: MLP, two fully connected hidden layers with dimension 128
- Neural Network activations: LeakyReLU, negative slope $= 0.01$
- Latent initial condition: $u(s, 0) = \exp(-(x - 2.5)^2 / 0.1)$.
- Latent noise processes: $\rho = 0.02$, $\ell = 0.1$, $\mathbf{R} = \mathrm{diag}(0.1^2)$.
- Latent discretisation: FEM, $C^0([0, 1])$ polynomial trial/test functions, Crank-Nicolson time discretisation, $dt = 0.02$.
- Optimiser: Adam, learning rate $= 0.001$.

To generate the data, we simulate the advection equation with $dt = 0.02$, observing every 10 time-steps for $\Delta_t = 0.2$ and $N = 200$. Latent dimensions are $n_u = 64$ and $n_x = 64$, with each image $28 \times 28$ pixels. These images are then flattened to vectors $\mathbf{y}_n \in [0, 1]^{n_y}$, with $n_y = 784$.

**KDV equation experiment.**

- Time-series length: $N = 100$.
- Input: $n_y = 1792$.
- Pseudo-observations: $n_x = 40$.
- Latent $n_u = 600$.
- Decoder: $p_\theta(\mathbf{y}_n | \mathbf{x}_n) = \mathrm{Bern}(\mu_\theta(\mathbf{x}_n))$

- $\mu_\theta(\cdot)$: MLP, two fully connected hidden layers with dimension 128
- Encoder: $q_\phi(\mathbf{x}_n|\mathbf{y}_n) = \mathcal{N}(\mu_\phi(\mathbf{y}_n), \sigma_\phi(\mathbf{y}_n))$
- $\mu_\phi(\cdot), \sigma_\phi(\cdot)$: MLP, two fully connected hidden layers with dimension 128
- Neural Network activations: LeakyReLU, negative slope $= 0.01$
- Latent initial condition: $u(s, 0) = \cos(\pi s)$.
- Latent noise processes: $\rho = 0.01$, $\ell = 0.2$, $\mathbf{R} = \mathrm{diag}(0.05^2)$.
- Latent discretisation: Petrov-Galerkin approach of Debussche & Printems (1999): $C^0([0, 2])$ polynomial trial functions, Crank-Nicolson time discretisation, $dt = 0.01$.
- Parameter prior: $p(\alpha) = \mathcal{N}(1.5, 0.3^2)$.
- Optimiser: Adam, learning rate $= 0.005$.

To generate the data we simulate the KDV equation with $dt = 0.01$, observing every timestep for $\Delta_t = 0.01$, and we take $N = 100$ observations $\mathbf{y}_n$, with $\mathbf{y}_n \in [0, 1]^{n_y}$. Each $\mathbf{y}_n$ is a flattened image of dimension $n_y = 64 \times 28 = 1792$, and we encode to pseudo-observations $\mathbf{x}_n$ of dimension $n_x = 40$. The latent state dimension is $n_u = 600$. Predictive MSE is calculated by encoding $\mathbf{y}_t, t = 0.1$ to $\mathbf{x}_t, t = 0.1$, sampling $\mathbf{x}_t$ from the generative model forward in time for 10 time-steps (up to $t = 0.2$), and computing the normalised MSE of the decoded samples $\hat{\mathbf{y}}_t$ compared to the ground truth.

**Linear decoding/encoding.** If a linear data generation is assumed from $\mathbf{x}_{1:N}$ to $\mathbf{y}_{1:N}$, then this structure can inform decoding. With a linear decoder of the form:

$$p_\mathbf{A}(\mathbf{y}|\mathbf{x}) = \mathcal{N}(\mathbf{A}\mathbf{x}, \eta^2 I).$$

In this case, we use the "inverted" linear decoder given by:

$$q_\mathbf{A}(\mathbf{x}|\mathbf{y}) = \mathcal{N}((\mathbf{A}^\top\mathbf{A})^{-1}\mathbf{A}^\top\mathbf{y}, \eta^2(\mathbf{A}^\top\mathbf{A})^{-1}).$$

By selecting the encoder appropriately, the space of parameterised variational distributions can be restricted to align with our beliefs about the data generation process.

## B  FULL VARIATIONAL FRAMEWORK

We derive the approximate ELBO for joint estimation of dynamic parameters $\mathbf{\Lambda}$, and autoencoder parameters $\phi, \theta$. We start by writing the evidence

$$p(\mathbf{y}_{1:N}) = \int p(\mathbf{u}_{1:N}, \mathbf{x}_{1:N}, \mathbf{\Lambda}, \mathbf{y}_{1:N}) \mathrm{d}\mathbf{x}_{1:N} \mathrm{d}\mathbf{u}_{1:N} \mathrm{d}\mathbf{\Lambda}.$$

We maximize $\log p(\mathbf{y}_{1:N})$ as

$$\log p(\mathbf{y}_{1:N}) = \log \int p(\mathbf{u}_{1:N}, \mathbf{x}_{1:N}, \mathbf{\Lambda}, \mathbf{y}_{1:N}) \mathrm{d}\mathbf{x}_{1:N} d\mathbf{u}_{1:N} \mathrm{d}\mathbf{\Lambda}$$

$$= \log \int \frac{p(\mathbf{u}_{1:N}, \mathbf{x}_{1:N}, \mathbf{\Lambda}, \mathbf{y}_{1:N})}{q(\mathbf{u}_{1:N}, \mathbf{x}_{1:N}, \mathbf{\Lambda}|\mathbf{y}_{1:N})} q(\mathbf{u}_{1:N}, \mathbf{x}_{1:N}, \mathbf{\Lambda}|\mathbf{y}_{1:N}) \mathrm{d}\mathbf{x}_{1:N} \mathrm{d}\mathbf{u}_{1:N} \mathrm{d}\mathbf{\Lambda}$$

$$\geq \int \log \left[ \frac{p(\mathbf{u}_{1:N}, \mathbf{x}_{1:N}, \mathbf{\Lambda}, \mathbf{y}_{1:N})}{q(\mathbf{u}_{1:N}, \mathbf{x}_{1:N}, \mathbf{\Lambda}|\mathbf{y}_{1:N})} \right] q(\mathbf{u}_{1:N}, \mathbf{x}_{1:N}, \mathbf{\Lambda}|\mathbf{y}_{1:N}) \mathrm{d}\mathbf{x}_{1:N} \mathrm{d}\mathbf{u}_{1:N} \mathrm{d}\mathbf{\Lambda}$$

$$= \mathrm{ELBO},$$

where the third line follows from the application of Jensen's inequality. Our generative model determines the factorisation of the joint distribution, given in equation 1:

$$p(\mathbf{u}_{1:N}, \mathbf{x}_{1:N}, \mathbf{\Lambda}, \mathbf{y}_{1:N}) = p(\mathbf{y}_{1:N}|\mathbf{x}_{1:N}, \mathbf{u}_{1:N}, \mathbf{\Lambda}) p(\mathbf{x}_{1:N}|\mathbf{u}_{1:N}, \mathbf{\Lambda}) p(\mathbf{u}_{1:N}|\mathbf{\Lambda}) p(\mathbf{\Lambda})$$

$$= p_\theta(\mathbf{y}_{1:N}|\mathbf{x}_{1:N}) p(\mathbf{x}_{1:N}|\mathbf{u}_{1:N}, \mathbf{\Lambda}) p(\mathbf{u}_{1:N}|\mathbf{\Lambda}) p(\mathbf{\Lambda}).$$

Next, we plug this factorised distribution into the ELBO and obtain

$$\mathrm{ELBO} = \int \log \left[ \frac{p_\theta(\mathbf{y}_{1:N}|\mathbf{x}_{1:N}) p(\mathbf{x}_{1:N}|\mathbf{u}_{1:N}, \mathbf{\Lambda}) p(\mathbf{u}_{1:N}|\mathbf{\Lambda}) p(\mathbf{\Lambda})}{q(\mathbf{u}_{1:N}, \mathbf{x}_{1:N}, \mathbf{\Lambda}|\mathbf{y}_{1:N})} \right]$$

$$\times q(\mathbf{u}_{1:N}, \mathbf{x}_{1:N}, \mathbf{\Lambda}|\mathbf{y}_{1:N}) \mathrm{d}\mathbf{x}_{1:N} \mathrm{d}\mathbf{u}_{1:N} \mathrm{d}\mathbf{\Lambda}.$$

The family of distributions which we use to approximate the posterior is described below. We assume a factorisation based on the model into variational encoding $q_\phi(\cdot)$, a full latent state posterior $q_\nu(\cdot)$, and the variational approximation to the parameter posterior $q_\lambda(\cdot)$:

$$q(\mathbf{u}_{1:N}, \mathbf{x}_{1:N}, \mathbf{\Lambda}|\mathbf{y}_{1:N}) = q_\nu(\mathbf{u}_{1:N}|\mathbf{x}_{1:N}, \mathbf{\Lambda})q_\lambda(\mathbf{\Lambda})q_\phi(\mathbf{x}_{1:N}|\mathbf{y}_{1:N}) \tag{6}$$

$$= q_\nu(\mathbf{u}_{1:N}|\mathbf{x}_{1:N}, \mathbf{\Lambda})q_\lambda(\mathbf{\Lambda})\prod_{n=1}^{N} q_\phi(\mathbf{x}_n|\mathbf{y}_n). \tag{7}$$

The second line demonstrates the amortized structure of the autoencoder, where the same encoding parameters are shared across datapoints. We can then substitute this expression into our ELBO and obtain

$$\text{ELBO} = \int \log \left[ \frac{p_\theta(\mathbf{y}_{1:N}|\mathbf{x}_{1:N})p(\mathbf{x}_{1:N}|\mathbf{u}_{1:N}, \mathbf{\Lambda})p(\mathbf{u}_{1:N}|\mathbf{\Lambda})p(\mathbf{\Lambda})}{q_\nu(\mathbf{u}_{1:N}|\mathbf{x}_{1:N}, \mathbf{\Lambda})q_\lambda(\mathbf{\Lambda})q_\phi(\mathbf{x}_{1:N}|\mathbf{y}_{1:N})} \right]$$
$$\times q(\mathbf{u}_{1:N}, \mathbf{x}_{1:N}, \mathbf{\Lambda}|\mathbf{y}_{1:N})\text{d}\mathbf{x}_{1:N}\text{d}\mathbf{u}_{1:N}\text{d}\mathbf{\Lambda}.$$

Assuming the variational posterior is the exact filtering posterior, i.e., $q_\nu(\mathbf{u}_{1:N}|\mathbf{x}_{1:N}, \mathbf{\Lambda}) = p(\mathbf{u}_{1:N}|\mathbf{x}_{1:N}, \mathbf{\Lambda})$ then applying Bayes rule

$$\frac{p(\mathbf{x}_{1:N}|\mathbf{u}_{1:N}, \mathbf{\Lambda})p(\mathbf{u}_{1:N}|\mathbf{\Lambda})}{q_\nu(\mathbf{u}_{1:N}|\mathbf{x}_{1:N}, \mathbf{\Lambda})} = p(\mathbf{x}_{1:N}|\mathbf{\Lambda})$$

leads to a simplification of ELBO in terms of the marginal likelihood $p(\mathbf{x}_{1:N}|\mathbf{\Lambda})$ of the state-space model. Substituting this expression leads to

$$\text{ELBO} = \int \log \left[ \frac{p_\theta(\mathbf{y}_{1:N}|\mathbf{x}_{1:N})p(\mathbf{x}_{1:N}|\mathbf{\Lambda})p(\mathbf{\Lambda})}{q_\phi(\mathbf{x}_{1:N}|\mathbf{y}_{1:N})q_\lambda(\mathbf{\Lambda})} \right]$$
$$\times q_\nu(\mathbf{u}_{1:N}|\mathbf{x}_{1:N}, \mathbf{\Lambda})q_\lambda(\mathbf{\Lambda})q_\phi(\mathbf{x}_{1:N}|\mathbf{y}_{1:N})\text{d}\mathbf{x}_{1:N}\text{d}\mathbf{u}_{1:N}\text{d}\mathbf{\Lambda}$$
$$= \int \log \left[ \frac{p_\theta(\mathbf{y}_{1:N}|\mathbf{x}_{1:N})}{q_\phi(\mathbf{x}_{1:N}|\mathbf{y}_{1:N})} \right] q_\phi(\mathbf{x}_{1:N}|\mathbf{y}_{1:N})\text{d}\mathbf{x}_{1:N}$$
$$+ \int \left[ \log p(\mathbf{x}_{1:N}|\mathbf{\Lambda}) + \log \frac{p(\mathbf{\Lambda})}{q_\lambda(\mathbf{\Lambda})} \right] q_\lambda(\mathbf{\Lambda})q_\phi(\mathbf{x}_{1:N}|\mathbf{y}_{1:N})\text{d}\mathbf{x}_{1:N}\text{d}\mathbf{\Lambda}$$
$$= \mathbb{E}_{q_\phi} \left[ \log \frac{p_\theta(\mathbf{y}_{1:N}|\mathbf{x}_{1:N})}{q_\phi(\mathbf{x}_{1:N}|\mathbf{y}_{1:N})} \right] + \mathbb{E}_{q_\phi} \left[ \mathbb{E}_{q_\lambda} \left[ \log p(\mathbf{x}_{1:N}|\mathbf{\Lambda}) + \log \frac{p(\mathbf{\Lambda})}{q_\lambda(\mathbf{\Lambda})} \right] \right],$$

Using a single MC sample from $q_\phi(\mathbf{x}_{1:N}|\mathbf{y}_{1:N})$ to approximate the expectation, we can write:

$$\mathcal{T}(\theta, \phi, \lambda) = \log \frac{p_\theta(\mathbf{y}_{1:N}|\mathbf{x}_{1:N})}{q_\phi(\mathbf{x}_{1:N}|\mathbf{y}_{1:N})} + \mathbb{E}_{q_\lambda} \left[ \log p(\mathbf{x}_{1:N}|\mathbf{\Lambda}) \right] - \mathcal{KL}(q_\lambda(\mathbf{\Lambda}), p(\mathbf{\Lambda})).$$

We can sample $q_\lambda(\mathbf{\Lambda})$ to approximate the expectation of $\log p(\mathbf{x}_{1:N}|\mathbf{\Lambda})$. Note that this requires the reparameterisation trick that is used for also used when sampling $\mathbf{x}_{1:N}$. This allows for backpropagation of errors through the sampling step. The KL-divergence can be calculated analytically for the case of Gaussian prior and posterior:

$$\mathcal{T}(\theta, \phi, \lambda) = \log \frac{p_\theta(\mathbf{y}_{1:N}|\mathbf{x}_{1:N})}{q_\phi(\mathbf{x}_{1:N}|\mathbf{y}_{1:N})} + \frac{1}{M} \sum_{i=1}^{M} \left[ \log p(\mathbf{x}_{1:N}|\mathbf{\Lambda}^{(i)}) \right] - \mathcal{KL}(q_\lambda(\mathbf{\Lambda}), p(\mathbf{\Lambda})),$$

and approximated via Monte-Carlo otherwise

$$\mathcal{T}(\theta, \phi, \lambda) = \log \frac{p_\theta(\mathbf{y}_{1:N}|\mathbf{x}_{1:N})}{q_\phi(\mathbf{x}_{1:N}|\mathbf{y}_{1:N})} + \frac{1}{M} \sum_{i=1}^{M} \left[ \log p(\mathbf{x}_{1:N}|\mathbf{\Lambda}^{(i)}) + \log p(\mathbf{\Lambda}^{(i)}) - \log q_\lambda(\mathbf{\Lambda}^{(i)}) \right].$$

## C  FURTHER DETAILS ON THE DYNAMIC MODEL

In this work we take the latent dynamical model to be a stochastic ODE or PDE. For an ODE this follows from a standard SDE (Särkkä & Solin, 2019; Øksendal, 2003), given by

$$\text{d}\mathbf{u} = f_\Lambda(\mathbf{u}, t; \Lambda)\text{d}t + \mathbf{L}(t)\text{d}\mathbf{W}(t),$$

where $\mathbf{u} := \mathbf{u}(t) \in \mathbb{R}^{n_u}$, $t \in [0, T]$, $f_\Lambda \colon \mathbb{R}^{n_u} \times [0, T] \to \mathbb{R}^{n_u}$, $\mathbf{L} \colon [0, T] \to \mathbb{R}^{n_u \times n_u}$. The noise process $\mathbf{W}(t)$ is a standard vector Brownian motion. The diffusion term $\mathbf{L}(t)$ can be used to describe any *a priori* correlation in the error process dimensions. As stated in the main text, this error process is taken to represent possibly misspecified/unknown physics, which may have been omitted when specifying the model. Discretisation with an explicit Euler-Maruyama scheme (Kloeden & Platen, 1992) gives,

$$\mathbf{u}_n = \mathbf{u}_{n-1} + \Delta_t f_{n-1}(\mathbf{u}_{n-1}; \Lambda) + \mathbf{L}_{n-1} \Delta \mathbf{W}_{n-1}, \quad \Delta \mathbf{W}_{n-1} \sim \mathcal{N}(\mathbf{0}, \Delta_t \mathbf{I}),$$

where $\mathbf{u}_n := \mathbf{u}(n\Delta_t)$, $f_n(\cdot; \Lambda) = f_\Lambda(\cdot, n\Delta_t)$, and so on. This gives a transition density

$$p(\mathbf{u}_n | \mathbf{u}_{n-1}, \Lambda) = \mathcal{N}(\mathbf{u}_{n-1} + \Delta_t f_{n-1}(\mathbf{u}_{n-1}; \Lambda), \Delta_t \mathbf{L}_{n-1} \mathbf{L}_{n-1}^\top),$$

defining a Markov model on the now discretised state vector $\mathbf{u}_n$. To align with the notation introduced in the main text, this gives:

$$p(\mathbf{u}_n | \mathbf{u}_{n-1}, \Lambda) = \mathcal{N}(\mathcal{M}(\mathbf{u}_{n-1}), \mathbf{Q}),$$

$$\mathcal{M}(\mathbf{u}_{n-1}) := \mathbf{u}_{n-1} + \Delta_t f_{n-1}(\mathbf{u}_{n-1}; \Lambda), \quad \mathbf{Q} := \Delta_t \mathbf{L}_{n-1} \mathbf{L}_{n-1}^\top.$$

Due to the structure of the STATFEM discretisation, the fully-discretised underlying model is of the same mathematical form as this ODE case. The difference lies in the dynamics being defined from either a PDE or ODE system. In common cases, a lower dimensional state vector, $\mathbf{u}_n$, typically results for the ODE case in comparison to the PDE case. For the PDE case, entries of the state vector $\mathbf{u}_n$ are coefficients of the finite element basis functions.

For the PDE case, the derivation is similar, with an additional step pre-time-discretisation to spatially discretise the system. This yields a method-of-lines approach (Schiesser, 1991). As in the main text, we consider a generic nonlinear PDE system of the form

$$\partial_t u + L_\Lambda u + F_\Lambda(u) = f + \dot{\xi}, \quad \dot{\xi} \sim \mathcal{GP}(0, \delta(t - t') \cdot k(\mathbf{s}, \mathbf{s}')), \tag{8}$$

where $u := u(\mathbf{s}, t)$, $\xi := \xi(\mathbf{s}, t)$, $f := f(\mathbf{s})$, $\mathbf{s} \in \Omega \subset \mathbb{R}^d$, and $t \in [0, T]$. The operators $L_\Lambda$ and $F_\Lambda(\cdot)$ are linear and nonlinear differential operators, respectively. The process $\dot{\xi}$ is the derivative of a function-valued Wiener process, whose increments are given by a Gaussian process with the covariance kernel $k(\cdot, \cdot)$. In our examples, we use the squared-exponential covariance function (Williams & Rasmussen, 2006)

$$k(\mathbf{s}, \mathbf{s}') = \rho^2 \exp\left(-\frac{\|\mathbf{s} - \mathbf{s}'\|_2^2}{2\ell^2}\right).$$

Hyperparameters $\{\rho, \ell\}$ are always assumed to be known, being set *a priori*. Further work investigating inference of these hyperparameters is of interest.

As stated in the main text we discretise the linear time-evolving PDE following the STATFEM as in Duffin et al. (2021), for which we refer to for the full details of this approach. In brief, we discretise spatially with finite elements (see, e.g., Brenner & Scott (2008); Thomée (2006), for standard references) then temporally via finite differences. We first multiply equation 8 with a sufficiently smooth test function $v \in V$, where $V$ is an appropriate function space (e.g. the $H_0^1(\Omega)$ Sobolev space (Evans, 2010)) and integrate over the domain $\Omega$ to give the weak form (Brenner & Scott, 2008)

$$\langle \partial_t u, v \rangle + \mathcal{A}_\Lambda(u, v) + \langle F_\Lambda(u), v \rangle = \langle f, v \rangle + \langle \dot{\xi}, v \rangle, \quad \forall v \in V.$$

Recall that $\mathcal{A}_\Lambda(\cdot, \cdot)$ is the bilinear form generated from the linear operator $L_\Lambda$, and

$$\langle f, g \rangle = \int_\Omega f(\mathbf{s}) g(\mathbf{s}) \, \mathrm{d}\mathbf{s},$$

the $L^2(\Omega)$ inner product.

Next we introduce a discrete approximation to the domain, $\Omega_h \subseteq \Omega$, having vertices $\{\mathbf{s}_j\}_{j=1}^{n_h}$. This is parameterised by $h$ which indicates the degree of mesh-refinement. We now introduce a finite-dimensional set of polynomial basis functions $\{\phi_j(\mathbf{s})\}_{j=1}^{n_u}$, such that $\phi_i(\mathbf{s}_j) = \delta_{ij}$. In this work these are exclusively the $C^0(\Omega)$ linear polynomial "hat" basis functions. This gives the finite-dimensional function space $V_h = \mathrm{span}\{\phi_j(\mathbf{s})\}_{j=1}^{n_u}$, which is the space we look for solutions

in. Next, we rewrite $u$ and $v$ in terms of these basis functions: $u_h(\mathbf{s}, t) = \sum_{j=1}^{n_u} u_j(t)\phi_j(\mathbf{s})$ and $v_h(\mathbf{s}, t) = \sum_{j=1}^{n_u} v_j(t)\phi_j(\mathbf{s})$. As the weak form must hold for all $v_h \in V_h$, this is equivalent to holding for all $\phi_j$. Thus, the weak form can now be rewritten in terms of this set of basis functions

$$\langle \partial_t u_h, \phi_j \rangle + \mathcal{A}_\Lambda(u_h, \phi_j) + \langle F_\Lambda(u_h), \phi_j \rangle = \langle f, \phi_j \rangle + \langle \dot{\xi}, \phi_j \rangle, \quad j = 1, \ldots, n_u.$$

Note that, in general, $u_h$ and $v_h$ do not necessarily have to be defined on the same function space, but as we use the linear basis functions in this work we stick with this here.

As stated in the main text, this is an SDE over the FEM coefficients $\mathbf{u}(t) = (u_1(t), \ldots, u_{n_u}(t))^\top$, given by

$$\mathbf{M}\frac{\mathrm{d}\mathbf{u}}{\mathrm{d}t} + \mathbf{A}\mathbf{u} + \mathcal{F}(\mathbf{u}) = \mathbf{b} + \dot{\boldsymbol{\xi}}, \quad \dot{\boldsymbol{\xi}}(t) \sim \mathcal{N}(\mathbf{0}, \delta(t - t') \cdot \mathbf{G})$$

where $\mathbf{M}_{ij} = \langle \phi_i, \phi_j \rangle$, $\mathbf{A}_{ij} = \mathcal{A}_\Lambda(\phi_i, \phi_j)$, $\mathcal{F}(\mathbf{u})_j = \langle F_\Lambda(u_h), \phi_j \rangle$, $\mathbf{b}_j = \langle f, \phi_j \rangle$, and $\mathbf{G}_{ij} = \langle \phi_i, \langle k(\cdot, \cdot), \phi_j \rangle \rangle$. Letting $\mathbf{G} = \mathbf{L}\mathbf{L}^\top$ we can then write this in the familiar notation as above

$$\mathbf{M}\mathrm{d}\mathbf{u} + \mathbf{A}\mathbf{u}\mathrm{d}t + \mathcal{F}(\mathbf{u})\mathrm{d}t = \mathbf{b}\mathrm{d}t + \mathbf{L}\,\mathrm{d}\mathbf{W}(t),$$

from which an Euler-Maruyama time discretisation gives

$$\mathbf{u}_n = (\mathbf{I} - \Delta_t \mathbf{M}^{-1}\mathbf{A})\mathbf{u}_{n-1} - \Delta_t \mathbf{M}^{-1}\mathcal{F}(\mathbf{u}_{n-1}) + \Delta_t \mathbf{M}^{-1}\mathbf{b} + \mathbf{M}^{-1}\mathbf{L}\Delta\mathbf{W}_{n-1},$$

where $\Delta\mathbf{W}_{n-1} \sim \mathcal{N}(0, \Delta_t \mathbf{I})$, eventually defining a transition model of the form

$$p_\Lambda(\mathbf{u}_n|\mathbf{u}_{n-1}) = \mathcal{N}\left((\mathbf{I} - \Delta_t \mathbf{M}^{-1}\mathbf{A})\mathbf{u}_{n-1} - \Delta_t \mathbf{M}^{-1}\mathcal{F}(\mathbf{u}^{n-1}) + \Delta_t \mathbf{M}^{-1}\mathbf{b}, \Delta_t \mathbf{M}^{-1}\mathbf{G}\mathbf{M}^{-\top}\right). \tag{9}$$

Note that also that the STATFEM methodology also allows for implicit discretisations which may be desirable for time-integrator stability. The transition equations for these approaches can be written out in closed form, yet although they give Markovian transition models, the resultant transition densities $p(\mathbf{u}_n|\mathbf{u}_{n-1}, \boldsymbol{\Lambda})$ are not necessarily Gaussian due to the nonlinear dynamics being applied to the current state $\mathbf{u}_n$. Letting $\mathbf{e}_{n-1} = \mathbf{L}\Delta\mathbf{W}_{n-1} \sim \mathcal{N}(\mathbf{0}, \Delta_t \mathbf{G})$, then the implicit Euler is

$$\mathbf{M}\left(\mathbf{u}_n - \mathbf{u}_{n-1}\right) + \Delta_t \mathbf{A}\mathbf{u}_n + \Delta_t \mathcal{F}(\mathbf{u}_n) + \Delta_t \mathbf{b} = \mathbf{e}_{n-1}, \tag{10}$$

and the Crank-Nicolson is

$$\mathbf{M}\left(\mathbf{u}_n - \mathbf{u}_{n-1}\right) + \Delta_t \mathbf{A}\mathbf{u}_{n-1/2} + \Delta_t \mathcal{F}(\mathbf{u}_{n-1/2}) + \Delta_t \mathbf{b} = \mathbf{e}_{n-1}, \tag{11}$$

where $\mathbf{u}_{n-1/2} = \left(\mathbf{u}_n + \mathbf{u}_{n-1}\right)/2$. Furthermore, to compute the marginal measure $p(\mathbf{u}_n|\boldsymbol{\Lambda})$ this also requires integrating over the previous solution $\mathbf{u}_{n-1}$; again due to nonlinear dynamics this will not necessarily be Gaussian.

In each of these cases, therefore, the transition equation is

$$\mathcal{M}(\mathbf{u}_n, \mathbf{u}_{n-1}) = \mathbf{e}_{n-1},$$

where we take, for the implicit Euler,

$$\mathcal{M}(\mathbf{u}_n, \mathbf{u}_{n-1}) = \mathbf{M}\left(\mathbf{u}_n - \mathbf{u}_{n-1}\right) + \Delta_t \mathbf{A}\mathbf{u}_n + \Delta_t \mathcal{F}(\mathbf{u}_n) + \Delta_t \mathbf{b}$$

and, for the Crank-Nicolson,

$$\mathcal{M}(\mathbf{u}_n, \mathbf{u}_{n-1}) = \mathbf{M}\left(\mathbf{u}_n - \mathbf{u}_{n-1}\right) + \Delta_t \mathbf{A}\mathbf{u}_{n-1/2} + \Delta_t \mathcal{F}(\mathbf{u}_{n-1/2}) + \Delta_t \mathbf{b}.$$

In practice due to conservative properties of the Crank-Nicolson discretisation we use this for all our models in this work.

Discretised solutions $\mathbf{u}_n$ are mapped at time $n$ to "pseudo-observations" via the observation process

$$\mathbf{x}_n = \mathbf{H}\mathbf{u}_n + \mathbf{r}_n, \quad \mathbf{r}_n \sim \mathcal{N}(\mathbf{0}, \mathbf{R}).$$

This observation process has the density $p_\nu(\mathbf{x}_n|\mathbf{u}_n)$ where $\nu = \{\mathbf{H}, \mathbf{R}\}$. As stated in the main text, the pseudo-observation operator $\mathbf{H}$ and observational covariance $\mathbf{R}$ are assumed known in this work. We typically use a diagonal covariance, setting $\mathbf{R} = \sigma^2 \mathbf{I}$. In the PDE case, for a given STATFEM discretisation as above, these pseudo-observations are assumed to be taken on a user-specified grid, given by $\mathbf{x}_{\mathrm{obs}} \in \mathbb{R}^{n_x}$. The pseudo-observation operator thus acts as an interpolant, such that

$$\mathbf{H}\mathbf{u}_n = \left[u_h(\mathbf{x}_{\mathrm{obs}}^1, n\Delta_t), u_h(\mathbf{x}_{\mathrm{obs}}^2, n\Delta_t), \ldots, u_h(\mathbf{x}_{\mathrm{obs}}^{n_x}, n\Delta_t)\right]^\top.$$

For the ODE case, we have worked with observation operators that extract relevant entries from the state vector. The pseudo-observations are mapped to high-dimensional observed data through a possibly nonlinear observation model which has the probability density $p_\theta(\mathbf{y}_n|\mathbf{x}_n)$. Recall that in this, $\theta$ are neural network parameters. This defines the decoding component of our model (see Figure 1).

**Nonlinear Filtering for Latent State Estimation.** To perform state inference given a set of pseudo-observations we use the ExKF. The ExKF constructs an approximate Gaussian posterior distribution via linearising about the nonlinear model $\mathcal{M}(\cdot)$. The action of the nonlinear $\mathcal{M}(\cdot)$ is approximated via tangent linear approximation. We will derive our filter in the general context of a nonlinear Gaussian SSM given by

$$\text{Transition:} \quad \mathcal{M}(\mathbf{u}_n, \mathbf{u}_{n-1}) = \mathbf{e}_{n-1}, \quad \mathbf{e}_n \sim \mathcal{N}(\mathbf{0}, \mathbf{Q}),$$
$$\text{Observation:} \quad \mathbf{x}_n = \mathbf{H}\mathbf{u}_n + \mathbf{r}_n, \quad \mathbf{r}_n \sim \mathcal{N}(\mathbf{0}, \mathbf{R}).$$

This allows for the use of implicit time-integrators and subsumes the derivation for the explicit case. We assume that at the previous timestep an approximate Gaussian posterior has been obtained, $p(\mathbf{u}_{n-1}|\mathbf{x}_{1:n-1}, \mathbf{\Lambda}) = \mathcal{N}(\mathbf{m}_{n-1}, \mathbf{C}_{n-1})$. For each $n$ the ExKF thus proceeds as:

1. Prediction step. Solve $\mathcal{M}(\hat{\mathbf{m}}_n, \mathbf{m}_{n-1}) = 0$ for $\hat{\mathbf{m}}_n$. Calculate the tangent linear covariance update:
$$\hat{\mathbf{C}}_n = \mathbf{J}_n^{-1} \left( \mathbf{J}_{n-1} \mathbf{C}_{n-1} \mathbf{J}_{n-1}^\top + \mathbf{Q} \right) \mathbf{J}_n^{-\top},$$
where $\mathbf{J}_n = \partial\mathcal{M}/\partial\mathbf{u}_n|_{\hat{\mathbf{m}}_n, \mathbf{m}_{n-1}}$ and $\mathbf{J}_{n-1} = \partial\mathcal{M}/\partial\mathbf{u}_{n-1}|_{\hat{\mathbf{m}}_n, \mathbf{m}_{n-1}}$.
This gives $p(\mathbf{u}_n|\mathbf{x}_{1:n-1}, \mathbf{\Lambda}) = \mathcal{N}(\hat{\mathbf{m}}_n, \hat{\mathbf{C}}_n)$.

2. Update step. Compute the posterior mean $\mathbf{m}_n$ and covariance $\mathbf{C}_n$:
$$\mathbf{m}_n = \hat{\mathbf{m}}_n + \hat{\mathbf{C}}_n \mathbf{H}^T (\mathbf{H}\hat{\mathbf{C}}_n \mathbf{H}^T + \mathbf{R})^{-1}(\mathbf{y}_n - \mathbf{H}\hat{\mathbf{m}}_n),$$
$$\mathbf{C}_n = \hat{\mathbf{C}}_n - \hat{\mathbf{C}}_n \mathbf{H}^T (\mathbf{H}\hat{\mathbf{C}}_n \mathbf{H}^T + \mathbf{R})^{-1} \mathbf{H}\hat{\mathbf{C}}_n.$$
This gives $p(\mathbf{u}_n|\mathbf{x}_{1:n}, \mathbf{\Lambda}) = \mathcal{N}(\mathbf{m}_n, \mathbf{C}_n)$.

The log-marginal likelihood can be calculated recursively, with each term of the log-likelihood computed after each prediction step:

$$\log p(\mathbf{x}_{1:N}|\mathbf{\Lambda}) = \sum_{n=2}^{N} \log p(\mathbf{x}_n|\mathbf{x}_{1:n-1}, \mathbf{\Lambda}),$$
$$p(\mathbf{x}_n|\mathbf{x}_{1:n-1}, \mathbf{\Lambda}) = \mathcal{N}(\mathbf{H}\hat{\mathbf{m}}_n, \mathbf{H}\hat{\mathbf{C}}_n \mathbf{H}^T + \mathbf{R}).$$

Note that although we focus on the ExKF other nonlinear filters could be used; two popular alternatives are the ensemble Kalman filter (Evensen, 2003) or, the particle filter (Doucet et al., 2000). For a linear dynamical model, such as the advection equation considered in Section 5.2, the ExKF reduces to the standard Kalman filter (Kalman, 1960).

As mentioned in the main text we can marginalise over the uncertainty in the encoder, via a Monte Carlo approximation:

$$p(\mathbf{u}_n|\mathbf{y}_{1:n}, \mathbf{\Lambda}) \approx \int p(\mathbf{u}_n|\mathbf{x}_{1:n}, \mathbf{\Lambda}) q_\phi(\mathbf{x}_{1:n}|\mathbf{y}_{1:n}) \, \mathrm{d}\mathbf{x}_{1:n} \tag{12}$$

$$\approx \frac{1}{M} \sum_{i=1}^{M} p(\mathbf{u}_n|\mathbf{x}_{1:n}^{(i)}, \mathbf{\Lambda}), \quad \mathbf{x}_{1:n}^{(i)} \sim q_\phi(\cdot|\mathbf{y}_{1:n}). \tag{13}$$

The intractable integral is approximated using samples from the encoder, which provides an approximate posterior in the form of a mixture of Gaussians distribution, where each $p(\mathbf{u}_n|\mathbf{x}_{1:n}^{(i)}, \mathbf{\Lambda}) = \mathcal{N}(\mathbf{m}_n^{(i)}, \mathbf{C}_n^{(i)})$. A similar marginalisation procedure can proceed over the parameters

$$p(\mathbf{u}_n|\mathbf{y}_{1:n}) \approx \int p(\mathbf{u}_n|\mathbf{x}_{1:n}, \mathbf{\Lambda}) q_\lambda(\mathbf{\Lambda}) q_\phi(\mathbf{x}_{1:n}|\mathbf{y}_{1:n}) \, \mathrm{d}\mathbf{\Lambda}\mathrm{d}\mathbf{x}_{1:n} \tag{14}$$

$$\approx \frac{1}{M_\mathbf{x} M_\mathbf{\Lambda}} \sum_{i=1}^{M_\mathbf{x}} \sum_{j=1}^{M_\mathbf{\Lambda}} p(\mathbf{u}_n|\mathbf{x}_{1:n}^{(i)}, \mathbf{\Lambda}^{(j)}), \quad \mathbf{x}_{1:n}^{(i)} \sim q_\phi(\cdot|\mathbf{y}_{1:n}), \ \mathbf{\Lambda}^{(j)} \sim q_\lambda(\cdot). \tag{15}$$

