# OpenReview forum: "$\Phi$-DVAE: Learning Physically Interpretable Representations with Nonlinear Filtering"
_ICLR.cc/2023/Conference — Submitted to ICLR 2023_

### Official Review · Reviewer_8UxT · 2022-10-17

**Confidence:** 4
**Correctness:** 4
**Technical Novelty And Significance:** 4
**Empirical Novelty And Significance:** 4
**Recommendation:** 8

**Clarity, Quality, Novelty And Reproducibility:**

The usage of the StatFEM within a dynamic VAE is a novel solution that allows to model more complex non-linear systems.

The presentation is clear an well motivated, but needs to be simplified to be more accessible to the average reader (see previous section).

**Strength And Weaknesses:**

_STRENGHTS_

1. This work in an interesting extension of the  StatFEM framework for modelling high dimensional data derived from an underlying low dimensional physical system with known dynamics.
2. While most of the previous work has focused on modelling simpler dynamics, the proposed model can handle highly nonlinear systems governed by ODEs or PDEs.
3. The added stochasticity of the StatFEM model allows to deal with potential model misspecifications.
4. The model performs well in the chosen experiments, outperforming or performing comparably to competing methods.

_WEAKNESSES_

1. The StatFEM framework of the dynamic model presented in section 3.1 is quite complex for the average reader in the ICLR community. In its current state i therefore think that this paper would not have the impact it could.
There are a number of improvements that the authors could do to make the paper more accessible, for example:

    1. A simple running example could be used throughout the paper to provide more intuition to the reader.
    2. It would be good to have the full derivation of the dynamic model for the lorenz-63 or advection PDE example (even in the appendix).
    3. You could add pseudo code for the method.

2. In the model you assume the emission distribution of the SSM to be known. How come did you do that? Couldn't you have learnt those parameters as well?

3. I was surprised to see that the VRNN is slightly better than your model in the Korteweg-de Vries PDE example. Can you provide more intuition on why this is the case? I would have expected a model like the VRNN, which has no inductive bias of physical knowledge, to have a hard time learning to model these compex dynamics.

**Summary Of The Paper:**

This work combines VAEs and nonlinear state space models to model high dimensional time series data of physical systems.

The proposed $\Phi$-DVAE uses a variational autoencoder to map the high dimensional data in a latent representation (latent observations) whose dynamics can be modeled by a non linear state space model derived from the ODEs or PDEs governing the physical system. Joint estimation of latent states and model parameters is obtained by maximizing the ELBO,  using the extended Kalman filter for state estimation of the SSM.


**Summary Of The Review:**

The paper is interesting and novel, but for me to argue for acceptance its exposition needs to be improved and simplified.

---

> ### Author Response · Authors · 2022-11-17
> **Response**
>
> We thank the reviewer for their comments on the novelty of our approach and performance of the model on the reported experiments.
>
> **The StatFEM framework of the dynamic model presented in section 3.1 is quite complex for the average reader
> in the ICLR community**
> We apologise for the lack of clarity. In order to address these points, we have included a full derivation of the model as the initial running example, presented in sec 3.1. As suggested, instead of including the generic nonlinear PDE, we now derive the method using KdV as the example equation.
>
> **In the model you assume the emission distribution of the SSM to be known. How come did you do that?
> Couldn’t you have learnt those parameters as well?**
> Indeed, these could have been learnt. As we have said for other reviewers, for the observation operator, given that we assume the latent dynamical system, information may be available on the relation between this latent dynamical system and the pseudo-observations. In this work we have chosen to incorporate this *a priori* information, and, following reviewer suggestions, we have noted this in Section 3.2. However, there may be little information available on how one should choose $\mathbf{R}$. Therefore, as we have stated to the other reviewer comments, additional work will focus on addressing the inference of these parameters.
>
> **I was surprised to see that the VRNN is slightly better than your model in the Korteweg-de Vries PDE example.**
> To produce a more relevant metric for comparison, we have included a 'predictive MSE'. Here, the model is given a single image frame which is encoded to a latent state, then we sample latent states forward in time, which are decoded and compared to the ground truth. The predictive MSE requires the model to have well-specified dynamic model to produce good reconstructions forward in time, and in this scenario, $\Phi$-DVAE outperforms the VRNN as expected  (see updated Figure 5, median predictive MSEs are 0.0415 and 0.0806 for $\Phi$-DVAE and VRNN respectively).

---

### Official Review · Reviewer_FFuK · 2022-10-20

**Confidence:** 3
**Correctness:** 3
**Technical Novelty And Significance:** 2
**Empirical Novelty And Significance:** 2
**Recommendation:** 5

**Clarity, Quality, Novelty And Reproducibility:**

### Clarity, quality

The method's description is clear, and the experiments are nicely reported.

### Novelty

As I mentioned in the previous form, the relation with other VAE-like sequential models is not overly clear, which prevents me from assessing the novelty. In my initial understanding, the major difference from previous methods is the use of the discretized nonlinear PDE for state transition. If this is correct, I would say the novelty remains somewhat marginal.



**Strength And Weaknesses:**

### Strengths

The paper is basically nicely written and easy to follow.

### Weaknesses

W1.
The relation to the related work is not clearly stated. Given the variety of VAE-like models with sequential structures, it is hard to evaluate the particular design choice of the proposed method unless it is manifested with a clear comparison to the most relevant models.

W2.
Although the authors motivate themselves by referring to the "traditional approaches" that "focus on well-defined observation operators whose functional forms are typically assumed to be known," (in the abstract), in my understanding, the proposed method also stands on an observation operator whose functional form is assumed to be known; the map from $u$ to $x$ is linear (Eq. (7)), and the map from $x$ to $y$ is supposed to be selected with prior knowledge (Section 3.3).

W3.
Experiments are quite artificial and no real data appears.


**Summary Of The Paper:**

The authors propose a VAE-like framework with a sequential latent variable following a (discretized) PDE. They test it for three different systems.

**Summary Of The Review:**

While the paper is nicely written basically, the description of the relation between the proposed methods and other VAE-like sequential models remains unclear. Also, the experiments are only with very artificial data and thus are not so strong.

---

> ### Author Response · Authors · 2022-11-17
> **Response**
>
> We thank the reviewer for their comments about the quality of the writing.
>
> **The relation to the related work is not clearly stated.**
> We have now included a discussion, at the end of the related work section, detailing the connection of the $\Phi$-DVAE to the main other approaches that are discussed in the related work.
>
> **The proposed method also stands on an observation operator whose functional form is assumed to be known.**
> Indeed, you are correct: the pseudo-observation operator is assumed to be known in this paper, and the mapping from the data to the pseudo-data is assumed to be unknown: we specify the relation between the pseudo-data and the latent dynamical system. As we have responded to the other reviewers, information may be available on the relation between the dynamical system and the observations, e.g., the observations may only be modulated by a single component of the dynamical system, which we have chosen to incorporate in our experiments. However we agree that loosening this requirement on the pseudo-observation operator would be of interest, and further work is planned to look into this.
>
> **Experiments are quite artificial and no real data appears.**
> We agree; future projects will focus on applications to experimental/field observations.

---

### Official Review · Reviewer_91wk · 2022-10-21

**Confidence:** 4
**Correctness:** 2
**Technical Novelty And Significance:** 2
**Empirical Novelty And Significance:** 2
**Recommendation:** 3

**Clarity, Quality, Novelty And Reproducibility:**

- **Clarity:** As I noted above, the paper is written very clearly! Below are some comments and please see the minor ones at the end of this question.
  - In the last paragraph of the intro, the authors mention that they address the problem of synthesising known physical models with diverse data streams. The formulation is a bit misleading as it can suggest that multiple different data streams can be given as input in the model, which is not the case in the experiments. Hence, I recommend rephrasing.
  - It would help if the authors would clearly specify in the model design that as the encoding process they see the encoder $p(x|y)$ and then the filtering problem is given by the data likelihood $p(x|u)$ in which they want to marginalise out the latent dynamic variables $u$. Hence, in section 3.2, they actually address the filtering problem. This is not clear from reading this section alone without referring to the supplementary document.
  - In general, from reading the text it is unclear whether the "latent encodings" refer to the embeddings of the VAE encoder $q(x|y)$ or the data likelihood $p(x|u)$.

- **Quality:** Concerning empirical evaluations, I have significant worries. I would be very happy if the authors can comment on these:
  - Concerning experiments 2 and 3; how meaningful is it to provide comparisons on a dataset generated via the model's own latent dynamics functional form? Don't we naturally expect the presented model to excel?
   - Concerning experiment 1; how realistic is the dataset? Similar approaches typically benchmark their methods on sequence data instead of vector field observations, so I'm not sure if this comparison and presented results translate into real-world scenarios. I would be happy if the authors can clarify the motivation for vector field observations and typical real-world scenarios involving such data.
   - Concerning experiment 2; KVAE assumes unknown pseudo-observation mappings, which would also prevent a fair comparison.
   - Concerning experiment 3; while the dataset comes from a PDE, comparison partners are auto-regressive methods. Is this a fair comparison? I highly encourage comparisons against PDE-based dynamical models.
   -  I am uncertain whether the obtained results support the claim that the rapid optimisation of the ELBO is a sign of the inductive bias forcing the learned representations of the data y to accord with the well-specified latent dynamic model. Indeed the ELBO converges extremely fast, however, MSE suggests that the reconstructions are still improving with an increasing number of iterations, suggesting that either the encoder or decoder or both parameters are still being optimised. As such, I do not see how the fast convergence of the ELBO is a sign that the encoding $q(x|y)$ matches the latent dynamics given by $u$. Similarly, I am missing support for the claim that the final trained model has learnt highly probable latent encodings.
   - For completeness, I wonder how the model would perform on the datasets that comparison partners are evaluated, e.g., bouncing balls.

- **Novelty:** The paper proposes a combination of existing methods (STATFEM and KVAE). I'm not sure if it the novelty is a strong side of the paper.

- **Reproducibility:** The experimental set-up, parameters and model design are described in the supplementary document. Hence, the paper is reproducible.


Minor comments:
- In this, **prior belief on** model parameters Λ are updated with data y to give a posterior distribution
- I disagree with the claim "In a typical scenario ... the observation model itself is typically assumed known". The whole literature of GPSSM and the recent latent neural ODE/SDE works do assume unknown observation mappings.
- neural networks are used to learn the unknown embedding **to/from** this lower dimensional space.
- parameters of the likelihood denoted **by**Statistical Finite Elements via Langevin Dynamics.
- "... might be generated only some observed dimensions of ... " is not clear to me.
- I would give eq8 right after eq4 as it is a part of the generative process rather than VI.
- Replacing in the final Gaussian SSM term ‘Observation’ with ‘Pseudo-Observation’ would be better to avoid confusion.
- The specifications of the probability density in sec3.3 seem irrelevant as they are task specific, hence, only needed to be mentioned/clarified in the given experiment set-up.
- For eq9, it would be more clear if the authors write out the true approximate posterior, and then add the claim that they assume that the variational posterior is the exact filtering posterior.


**Strength And Weaknesses:**

Strengths:
- The paper is very well-written. Most concepts are communicated clearly and the related work section was thorough.
- It touches upon a timely topic, i.e., latent differential equation systems with physical information.

Weaknesses:
- Both the pseudo-observation operator and the noise covariance are assumed to be known in this work. Isn't this too strong of an assumption?
- Since the model and inference heavily resemble KVAE (Fraccaro et al., 2017), the differences should be discussed. In my understanding, the linear dynamical system formulation in KVAE is generalized here. Prior dynamics parameters $\Lambda$ also resemble the coefficients of the linear transition model, i.e., $A$, and $b$ in KVAE; yet, the functional form of the transitions is assumed to be known here (with unknown parameters). Anything else?
 - Certain claims are not well-supported:
    - The authors mention that they form a ‘physics-informed prior on the latent space.’ However, I would argue that the latent space is the exact physical model rather than just a prior on it. They have argued that the noise term captures model misspecification, however, none of the experiments truly supports this claim: all of your experiments are with the true underlying physical model (with known/partially known or unknown parameters) of the system.
    - The following claim made is not well supported: 'this explicit likelihood is introduced to obtain a well-defined formulation for embedding’. The learned embedding for $x$ is given by the encoder $p(x|y)$. From the formulation, it is not immediately clear how eq7 affects the encoder in order to have ‘well-defined embeddings’. Based on the loss I see that we have a marginal likelihood term for $p(x|\lambda)$ in which we use the data likelihood term defined here. But at the moment I do not see in what way this would affect the variational encoding distribution $q(x|y)$. This should be (at least empirically) demonstrated.
- I have major concerns about the experiments, please see the "quality" point in the next question.

Questions:
- I'm not able to fully follow between eq5 and sec3.2. Is there any novelty here or is it an adaptation of STATFEM into this framework?



**Summary Of The Paper:**

The paper combines standard VAE with filter algorithms (for example extended Kalman filter). The VAE is used to encode and decode from observation space to the latent space (in the paper often referred to as pseudo-observation space). While the filter algorithm is used to estimate the marginal log likelihood over the latent physics process introduced via additional data likelihood term in the model. The parameters of the model are trained jointly using variational inference (ELBO). The model is tested on video and velocity field datasets. The model obtains similar or better performance compared to previous works in the field w.r.t MSE.

**Summary Of The Review:**

Overall the paper is very well-written and places itself nicely among the existing work. Acknowledging this is a subjective judgment, I find the proposed approach incremental as (i) many latent ODE/SDE/PDE works are already proposed, and (ii) the model does not achieve groundbreaking results on very interesting datasets. I also think knowing the functional form of the dynamics as well as the pseudo-observation operator and the noise covariance is slightly too restrictive. Certain claims are not well-supported (see "Weaknesses" question). I also have concerns about the datasets, comparison partners. and fairness of the comparisons. I recommend a reject but would be happy to update my review based on the author reply.

---

> ### Author Response · Authors · 2022-11-17
> **Major response**
>
> We thank the reviewer for their comments about the writing of the paper, and acknowledgement of the relevance of the problem we address.
>
> **Both the pseudo-observation operator and the noise covariance are assumed to be known in this work.**
> With respect, on the observation operator, given that we stipulate the latent dynamical system, information may be available on the relation between this latent dynamical system and the (pseudo-) observations. As we have updated in the main text, in Section 3.2, perhaps only a subset of the dynamical system components is generating the data. In our case, given that we stipulate the latent encoding dimension we believe it is reasonable to impose the additional constraint that the pseudo-observations are taken from an evenly spaced set of gridpoints, which align with the chosen PDE discretisation.
>
> In Section 5.1, we also know that the first component of the Lorenz system is the component which generates the velocity fields, so this information is incorporated in the dynamical system specification.
>
> On the latent noise covariance, yes, this would ideally be estimated. However, to demonstrate the method it was considered that no variance hyperparameters were to be estimated in the method (this includes the GP hyperparameters). We have now updated the text in order to reflect these modelling choices, and plan to investigate these further in future work.</strong>
>
> **Since the model and inference heavily resemble KVAE (Fraccaro et al., 2017), the differences should be discussed.**
> We agree that differences should be discussed. Following reviewers’ suggestions, we have now updated the main text to include a set of comparisons in the related work section, going through DVAE approaches, deep SSMs, and various physics-informed methods. Regarding the KVAE in particular, your understanding is correct, but we also include the generalisation to embed PDEs, and not just linear ODEs (as an equivalent derivation for KVAE would come from). The functional form is known up to unknown parameters, but the key thing here is that we place a prior over these unknown parameters and form posterior estimates of them, rather than purely learning coefficients of a linear system (as in the KVAE).
>
> **I would argue that the latent space is the exact physical model rather than just a prior on it.**
> The 'prior' over the latent space we refer to is $p(x)$. For standard VAE this is often chosen as a zero-mean isotropic Gaussian. With our method, this distribution $p(x)$ is a marginal-log-likelihood, as given by the extended Kalman filter which is constructed based on the assumed physics model. We would agree, that if we specified the latent transition model as deterministic, and no initial condition covariance, then the problem would be specified exactly, then the problem would become a simple autoencoding regression task. However, since we include misspecification, and can allow priors over dynamic parameters, we argue that we are placing a prior over the latent space, which we are informing based on the assumed physical model driving the transitions.
>
> **I do not see in what way this would affect the variational encoding distribution**
> The loss as seen in Eq. 10 inlcudes the marginal-likelihood p(x) term as you mention. This expectation of this term is taken with respect to the encoding distribution: $E_{q_{\phi}} [p(x)]$. This means the encoding parameters are updated to not only provide good reconstructions (first term of the loss), but also to maximise the marginal-likelihood p(x) which is based on the underlying dynamic model.
>
> **Is there any novelty here or is it an adaptation of statFEM into this framework?**
> Correct, we are just presenting the statFEM within this framework.
>
> **The formulation is a bit misleading...**
> Apologies for this, we have now updated the text to reflect this more appropriately.
>
> **It would help if the authors would clearly specify in the model design**
> Apologies for the opaque description. We have now updated Section 3.2 to include a note that we are addressing the filtering problem, and that the pseudo-data are generated via the encoding.
>
> **Unclear whether the "latent encodings" refer to the embeddings of the VAE encoder or the data likelihood**
> Apologies. We have now removed the density that was given in the last paragraph of Section 5.2, in order to avoid confusion when discussing the encodings.

---

> ### Author Response · Authors · 2022-11-17
> **Quality and Minor comments**
>
> **Quality**
> The following list contains our responses to the quality section of this review.
>
> * Please see our general response.
> * Many applications involving fluids involve the technique of particle image velocimetry (PIV) where particles are placed in a flow, and images taken and velocity fields reconstructed. These scenarios produce data of velocity fields throughout time, and in the Lorenz example, we observe this velocity field over a 2D grid through time.
> * The pseudo-observation operator for KVAE is learned, whereas we specify this. As part of specifying a particular dynamical system, we decide which parts of the latent states are unobserved, and set the pseudo-observation operator accordingly, e.g. a differential equation describes position and velocity which comprises the latent state, we assume as part of the modelling assumption we only observe the position. The encoding then learns the mapping from the data $y$ to the observed part of the latent state $x$.
> * With respect, in our opinion, these comparisons are fair. Once discretised, the continuous-time differential equation on the latent space becomes an autoregressive process of order one.
> * We agree that these claims are not supported by the results we have presented. We have removed these claims and replaced this with the following. **The ELBO for $\Phi$-DVAE is rapidly  optimised in comparison to the KVAE models, and is greater by the end of training. The trained $\Phi$-DVAE gives better evidence for the data (greater ELBO), whilst providing more accurate reconstructions (lower MSE).**
> * Indeed this would be of interest, particularly as we could encode a latent differential equation of motion. This will be an avenue of further work.
>
> **Minor Comments**
> The following list contains our responses to the minor comments section of this review.
> * Done
> * We agree that this was misleading. We have now clarified the text to refer to classical data assimilation tasks, which are the settings in which the observation operator is known.
> * Fixed.
> * Fixed.
> * Apologies for the lack of clarity. We have now updated the text to contain an example which illustrates how the dimensionalities may relate to one another (e.g., single components of a latent coupled differential equation generating the observations).
> * Indeed. The main text has been updated with this change.
> * Done.
> * Apologies for the superfluous material. We have now removed these densities, reserving them to only be mentioned in the experiments.
> * Agreed. We have now done fixed this in the main text, aligning with other reviewer comments.

---

> ### Author Response · Authors · 2022-11-30
> **Please let us know if you have more questions**
>
> Dear reviewer,
>
> We would appreciate if you could let us know if you have any more concerns or comments that are not addressed. If not, we would like you to reconsider recommendation for our work as a result of the rebuttal and increase your score if you are satisfied with our answers. Please let us know if you have any more questions, we are happy to discuss.
>
> Thanks,

---

### Official Review · Reviewer_66rK · 2022-10-25

**Confidence:** 2
**Correctness:** 3
**Technical Novelty And Significance:** 3
**Empirical Novelty And Significance:** 2
**Recommendation:** 5

**Clarity, Quality, Novelty And Reproducibility:**

### Clarity

While the paper is **overall well-written**, **some additional clarifications should be made** regarding the description of some components of the model and the derivation of the ELBO (cf. previous section of the review).

### Quality

Besides the aforementioned points, there are a few typos and formatting issues.
 - Section 2 (p. 2): "The Kalman variational autoencoder (KVAE) Fraccaro et al. (2017)" -> The Kalman variational autoencoder (KVAE) of Fraccaro et al. (2017).
 - Section 5 (p. 6): "Our experiment setup is thus mimics" -> Our experiment setup thus mimics
 - Section 3.3 and 4: functions definitions (for $\mu_{\theta}$) should use the `\colon` command rather than `:`.
 - The quality of of all plots in figures should be improved in order to be readable without color information.

### Novelty

Both **the tackled problem and the proposed model** are novel, even though their significance remains unclear given the current experimental results (cf. previous section of the review).

### Reproducibility

While the code is not provided by the authors at submission time, they indicate that they will release it upon publication, which should be checked to ensure proper reproducibility. Besides this, the paper and appendix **contain sufficient implementation details to hopefully replicate the results**.

**Strength And Weaknesses:**

### Strengths

To the best of my knowledge, this paper tackles the **relevant problem** of data assimilation and parameter estimation in the ML x physics community. In this regard, it is **well motivated** as it is designed to operate not directly on the true physical state like many prior approaches, but on an indirect observation of this state whose specifications are unknown. This setting, as well as the designed model, **are novel in the ML x physics domain**. Models and results are, for the most part, **clearly presented and well designed**, with explicit motivations for each of the model components accounting for known and unknown information in the studied phenomenon. **Encouraging experimental results** show the capacity of the model to correctly reconstruct the physical system.

### Weaknesses

There are two weaknesses that limit, to the best of my understanding, the significance of the presented contributions.

Most importantly, **the empirical evaluation of the model does not suffice to properly evaluate the contributions**, because of three main issues.
 - There is a **lack of relevant baselines** in the comparison. Section 5.1 includes no comparison, hence results can hardly be used to draw conclusions. Section 5.2 only considers KVAE, and Section 5.3 only VRNN, GP-VAE and a weak baseline (a non-dynamic VAE). This is insufficient to draw informed conclusions on the relevance of the proposed model. To the very least, VRNN, GP-VAE and KVAE should be included in all experiments. Moreover, other recent baselines could be considered, e.g. Karl et al. (2017, cited in the paper), Krishnan et al. (2017), Li et al. (2018), Yıldız et al. (2019, cited in the paper), de Bézenac et al. (2020), Lu et al. (2020, cited in the paper), to cite probabilistic methods only; note that many ODE/PDE baselines may be considered as well, cf. the Minor Issues section. Note that I am not asking for all these methods to be included but I would expect some of them to be in the paper; alternatively, I would be interested in a discussion from the authors about their relevance in our context.
 - The experiments **fail to prove the relevance of the proposed model w.r.t. the state of the art**. VRNN outperforms the proposed model in reconstruction MSE, and it appears that increasing the dimensionality of KVAE could outperform it as well -- higher values than $n_x = 64$ should be tested. In contrast with the proposed model which requires strong prior knowledge on the physical phenomenon, these baselines are generic and they would be expected to underperform, which is not the case.
 - Even though motivated in Section 3, **main model components are not properly justified with experimental results**. As highlighted in Section 3.1, the introduced dynamics model is stochastic to circumvent uncertainties in the state estimation, but this property is never assessed in the experiments, especially as all considered differential equations are in practice studied in their deterministic version. Moreover, the benefits of adding prior knowledge and inferring the true parameters of the system are not demonstrated in the experiments either: this could be solved by including an ablation study.

Secondly, there are **some clarity issues in the description of the model and its training**, in particular for the non-experts in physical data assimilation like in the ICLR audience. The discretization and implementation of the dynamics model in Section 3.1 are obscure and could benefit from more intuitive explanations. Moreover, the marginalization over $\mathbf{u}$ in Section 4 (paragraph "Nonlinear Filtering") would require further explanations. Finally, **the derivation of the ELBO in Section 4 should be clarified**. Indeed, the proof in Appendix B involves the assumption that the variational posterior is the exact filtering posterior, which is neither discussed nor apparent elsewhere in the paper.

Krishnan et al. Structured Inference Networks for Nonlinear State Space Models. AAAI 2017.\
Li et al. Disentangled Sequential Autoencoder. ICML 2018.\
de Bézenac et al. Normalizing Kalman Filters for Multivariate Time Series Analysis. NeurIPS 2020.

### Minor Issues

 - By design, the proposed model cannot handle non-Markovian data, as the latent dynamics model is a first-order differential equation and the encoder $q_{\phi}$ is factorized as $q_\phi (\mathbf{x} \mid \mathbf{y}) = \prod_n q_\phi (\mathbf{x}_n \mid \mathbf{y}_n)$. This is a limitation of the model that could have been easily avoided like in many deep SSM using filtering or smoothing strategies -- cf. Fraccaro (2018, Section 3.3).
 - The related work misses two relevant lines of work.
   - Some aspects of ML for physics were not addressed. One could refer for examples to the introduction and related work of Yin et al. (2021) containing missing references for parameter estimation and incorporation of physical priors in ML systems.
   - Since the paper deals with video-like input data, it would be beneficial to discuss the relations of the model with existing VAE-based approaches for video modeling, which have already been approached with latent and state-space methods. Cf. for instance the related work of Wu et al. (2021).

Yin et al. Augmenting Physical Models with Deep Networks for Complex Dynamics Forecasting. ICLR 2021.\
Wu et al. Greedy Hierarchical Variational Autoencoders for Large-Scale Video Prediction. CVPR 2021.

**Summary Of The Paper:**

The paper proposes a Bayesian, probabilistic approach to data assimilation and parameter estimation for physical systems with unknown observation process. It takes the form of a dynamic VAE integrating in its temporal model the parameterized differential equation that drives the physical phenomenon, allowing the model trained with an ELBO to jointly estimate the unknown parameters and the physical state. The empirical performance of the model is then assessed on three differential equations where its ability to reconstruct the signal and estimate its parameters are demonstrated.

**Summary Of The Review:**

This paper tackles a well motivated problem and introduces a well designed model to solve it, by integrating physical dynamics prior into a Bayesian, VAE-based model. However, I have concerns regarding the experimental results which I believe are not sufficient to show the relevance of the proposed model w.r.t. prior work, and some clarity issues should be fixed to improve the quality and accessibility of the paper to the ICLR audience. Therefore, I find this paper to lie below the acceptance threshold.

Nonetheless, I believe that my concerns can be addressed during the discussion phase with the authors and the other reviewers, and would be glad to increase my score in this case.

---

> ### Author Response · Authors · 2022-11-17
> **Rebuttal**
>
> We thank the reviewer for their comments outlining the relevance of the problem and novelty of the paper, as well as the remark on encouraging results.
>
> **lack of relevant baselines**
> We have chosen to separate the comparisons into linear and nonlinear. Hence for the linear advection example we compare with KVAE, which models transitions with a linear Gaussian SSM. For the nonlinear KdV example, we compare with the DVAE models allow for nonlinear transitions. These choices have been made to provide as best as possible a fair comparison.
>
> Indeed, we agree that a more thorough comparison to the literature would be of interest. To this end, we have added in a comparison section, as other reviewers have also suggested, at the end of the related work.
>
> **Fail to prove the relevance of the proposed model w.r.t. the state of the art**
>
> We note that since we are in the dynamical setting, the reconstruction error may not be the most informative metric about the performance of an algorithm that is capable of prediction (albeit important). To address these points, we have added two main experiments: (1) KVAE comparison with higher dimensions (reconstruction error), (2) Comparison of *predictive* MSEs of $\Phi$-DVAE vs. VRNN. Therefore to assess the advantage of our method against VRNN, we look at predictive MSEs.
>
> For comparison against KVAE, we have increased the dimension of latent space to greater values ($n_x=128$ and $n_x=256$) above that of our $\Phi$-DVAE model (with $n_x=64$). Here we still outperform KVAE, as expected, since we are specifying the form of latent transitions.
>
> For the normalised MSE comparison with $\Phi$-DVAE and VRNN, both models have low reconstruction MSE ($\Phi$-DVAE 2.4\%, VRNN 1\%) to produce visually similar results. To produce a more relevant metric for comparison, we have included a 'predictive MSE'. Here, the model is given a single image frame which is encoded to a latent state, then we sample latent states forward in time, which are decoded and compared to the ground truth. The predictive MSE requires the model to have well-specified dynamic model to produce good reconstructions forward in time, and in this scenario, $\Phi$-DVAE outperforms the VRNN as expected (see updated Figure 5, median predictive MSEs are 0.0415 and 0.0806 for $\Phi$-DVAE and VRNN respectively).
>
>
> **Main model components are not properly justified with experimental results.**
> We agree with this point. The inclusion of the StatFEM misspecification has not been explored through experimentation. We have chosen to develop the method including the statFEM misspecification with the aim of being robust to model misspecification when applying $\Phi$-DVAE to experimental/field data, where the model is inherently imperfect. In future work we plan to explore this further. We also agree an ablation study would be of interest, to determine the affect of imposing prior knowledge on parameter estimation; this is also planned future work.
>
> **Some clarity issues in the description of the model and its training**
> We apologise for this lack of clarity. Following reviewer suggestions, we have reworked the derivation of the statFEM component in the main text to follow the KdV equation as a running example, and have included some additional discussions through the derivation. We have removed more technical jargon and have tried to focus more on intuitive concepts.
>
> **The derivation of the ELBO in Section 4 should be clarified**
> We have added additional discussion into the text on the computation of the marginal likelihood, in Section~4, and have noted that we use the true filtering posterior in the derivation of ELBO. This is approximated with extended Kalman filter (ExKF) posterior in practice. For the ELBO, we have padded out the derivation in a similar fashion to Fraccaro et. al. (2017).
>
> **By design, the proposed model cannot handle non-Markovian data**
> Again, we apologise for the lack of clarity. Although the order of the time derivative is first-order, in this work we make no assumptions on the degree of time-derivative taken. Higher-order time-derivatives could be incorporated via splitting the governing DE into a system of coupled first-order equations, allowing for our model structure to be used. Whilst the deep SSM methodologies do allow for generic transitions, this is traded off against the interpretability of the physics that one has with $\Phi$-DVAE.
>
> **The related work misses two relevant lines of work**
> Apologies for these omissions. We have now taken the relevant references from these works and have added them into the main text. Please see the updated “related work” section for these additions.
>
> **Clarity**
> See section above for response.
>
> **Quality**
> Apologies for these omissions – these should now have been fixed. We have also adjusted the plots to readable in a grayscale format.

---

> > ### Comment · Reviewer_66rK · 2022-11-20
> > **Acknowledgement**
> >
> > I would like to thank the authors for their response, which I carefully read together with the other reviews and responses. I appreciate the introduced clarifications on the existing content and experiments. Nonetheless, many of my initial concerns still stand, also supported by other reviews: lack of baselines, unclear significance, absence of justification of model components.
> >
> > I still hesitate regarding my recommendation. The critical point that needs to be addressed is whether the paper's contribution is significant enough in the domain of probabilistic sequential models in which it is grounded, w.r.t. possibly insufficient experiments and restrictive model design. Therefore, I look forward to discussing with the other reviewers on this matter before taking a stand.
> >
> > NB: Another question is the significance of the paper is the ML x Physics domain, which is another matter in in which I am less qualified. But given the current orientation of the paper, I think that the merits of $\Phi$-DVAE should rather be discussed as a probabilistic sequential model.

---

> > > ### Author Response · Authors · 2022-11-28
> > > **Thanks for your response!**
> > >
> > > We thank the reviewer for their response. We would like to clarify a few points regarding their concerns below:
> > >
> > > **lack of baselines:** We note that our methodology, as applied to physical systems in mind, does not have a standard competitor. Competitors like KVAE, VRNN (and other neural differential equation based possible competitors) would not give any more information than what we already have: All these models learn (essentially) an uninterpretable latent space and may or may not outperform our model in their prediction. We stress that the real importance of our work is to find a way to relate a sequence of data points (video frames, velocity fields) to the actual governing physics. Doing so rids us of the need of specifying very flexible neural network based latent spaces and we get latent representations that are fully interpretable as the solutions of associated PDEs. We also note that this allows a lot of further follow up work: the PDE-related numerical methods or theoretical tools can be applied to understand the properties of the method.
> > >
> > > **unclear significance:** We would like to argue that there is significant value in what we do: Instead of using black-box schemes and showing a big table of predictive results that are outperforming other black-box models, we try to use the white-box nature of physical models that have been developed in the last few centuries. We aim at embedding this knowledge into representation learning in a principled manner. We understand there is a lot to explore and we would like it to be recognized that this is not an easy problem to solve because of several statistical challenges associated with it. We could even argue that a black-box algorithm that can learn to predict any sequence is almost a solved problem but our approach is what will provide transparency and clarity for these problems.
> > >
> > > **absence of justification of model components:** Our latent space model aims at capturing the latent physics, while “pseudo-observations” are specifying the dimensions of the latent space which could be driving “real-observations”. A good example is the Lorenz 63 model: In data-assimilation literature, this is a standard example where it is assumed that only the first dimension of it is observed. However, this first dimension, _in the real world_, would be observed as a velocity field which is the missing bit from the vast literature we provided as a part of our model.
> > >
> > > **The critical point that needs to be addressed is whether the paper’s contribution is significant enough in the domain of probabilistic sequential models in which it is grounded, w.r.t. possibly insufficient experiments and restrictive model design.**
> > >
> > > We appreciate this point - but note that we do not fundamentally aim at proposing a new algorithm for probabilistic sequential modelling - nor a new filtering methodology. Despite this, our algorithm has some significant novel aspects: (1) KVAE type models did not have meaningful parameters, instead they learn full matrices (which could result in unstable latent systems as there is no intrinsic mechanism controlling the eigenvalues of the dynamical system). We restrict our model class and provide a parameter estimation scheme despite the model not being a standard state-space model which is a methodology that didn’t exist before. (2) The integration of prior knowledge on the parameters, which is again, a straightforward but undone part of these models.
> > >
> > > We would like to kindly ask the reviewer to reevaluate their rating for our paper. We do believe that this is a significant first step to what will become standard in physics-based ML works: The white nature of existing physical systems _has to be integrated_ into deep learning based physics methods to ensure reliability, predictability of their behaviour, and stability.

---

> > > > ### Comment · Reviewer_66rK · 2022-12-02
> > > > **Positioning Issue**
> > > >
> > > > I would like to thank you for your follow-up response which clarified the main contribution that you would like to highlight. I completely agree that, to properly operate on physical data, it is highly relevant to augment standard models such as dynamical VAEs with some knowledge of the true system dynamics.
> > > >
> > > > However, as also pointed out by the other reviewers, the proposed approach is white-box as it directly integrates the whole training dynamics. This is not a problem per se but raises a positioning issue: the paper should then be approached from a physical perspective rather than from a probabilistic sequential modeling perspective. This is in line with your argument stating that adding more baselines from the probabilistic sequential modeling community would not change much the conclusions.
> > > >
> > > > I did not review this paper from an ML x Physics perspective because it was not positioned in its area, both in the ICLR reviewing system and in the paper itself. Given this discussion, I think that the positioning of the paper should be changed in this direction for a better assessment of the highlighted contribution. In this regard, I believe that there exists baselines and ablations to consider, or at least related work to discuss (cf. for example references in my review), on this type of systems integrating direct knowledge about the studied systems.
> > > >
> > > > I would finally like to note that my assessment could have been different if some model parts were more supported by the experiments. In particular, from my review:
> > > > > As highlighted in Section 3.1, the introduced dynamics model is stochastic to circumvent uncertainties in the state estimation, but this property is never assessed in the experiments, especially as all considered differential equations are in practice studied in their deterministic version.
> > > >
> > > > Proving the ability of the proposed model to handle SDEs would have been a significant contribution in the domain probabilistic sequential modeling, as this is still a challenging task for such methods, to the best of my knowledge.
> > > >
> > > > Therefore, I maintain my initial recommendation and would like to recommend to shift at least partially the positioning of the paper towards the ML x Physics area where this contribution could be properly evaluated.

---

> > > > > ### Author Response · Authors · 2022-12-04
> > > > > **Thanks**
> > > > >
> > > > > Thanks for your thoughtful response. We very much appreciate your points and we would like to add a few further points to the discussion.
> > > > >
> > > > > As you correctly point out in your positioning issue, our paper lies right in the middle of two distinct approaches: On one hand, probabilistic sequential models (state-space models, stochastic filtering), on the other, so-called physics-informed ML. As such, we appreciate informed reviewer input from both communities.
> > > > >
> > > > > Our approach aimed at providing a generic representation learning framework for unstructured data driven by some physical process powered by developments in probabilistic modelling, e.g., stochastic (Kalman, particle, ensemble etc.) filtering, parameter estimation schemes, and variational autoencoders. As you'd appreciate, the correctness of such an approach can only be properly evaluated by knowledgable assessments from both communities (if this was just reviewed by the physics-ML community, the potential of this work may not be fully appreciated).
> > > > >
> > > > > You could also see from our algorithmic framework that a lot of the pieces can be moved, state-space (statFEM) models can be changed w.r.t. physics (or chemistry, biology etc), inference (filtering) framework is also plug-and-play (e.g. particle, ensemble, sigma-point filtering is possible) w.r.t. to the intractability of the underlying system, the VAE architecture can be made more intuitive by the use of tailored architecture to specific types of data.
> > > > >
> > > > > All in all, we'd like you to evaluate the paper from a probabilistic (stochastic) modelling perspective. In this setting, we argue that we provide something many people can build on to model unstructured (physical) data which is an important emerging frontier in the physical, natural and medical sciences. It is in this context that we ask you to reevaluate our paper.
> > > > >
> > > > > We thank you for your valuable input!

---

### Comment · Area_Chair_oPAE · 2022-11-15
**Please engage before the author-reviewer discussion closes**

Dear authors and reviewers,

The first phase of the discussion period is about to close on November 18.

For authors, please make sure to submit your rebuttal by the deadline. Leave some time for the reviewers to read it and respond while you are still allowed to further engage with them. Interactions between authors and reviewers are very important for the quality of the review process, so please make sure to engage.

For reviewers, please try to acknowledge and respond to the authors' rebuttal while the discussion period is still open for them to further interact with you.

Thank you for your participation in the review process!

Best,
The AC

---

### Author Response · Authors · 2022-11-17
**General response to reviewers**

We thank the reviewers for their thoughtful feedback. We have since revised the paper, and have included point-by-point responses to the comments received. Hence, we have uploaded a revised version of the manuscript, containing these revisions. Among other considerations, we have addressed typos throughout the text and have cleaned the presentation, and, as such, have removed superfluous content. Following, referee suggestions we have also updated the Figures, to be readable without colour information.

We have responded individually to the reviewers comments. However, we note here that a general criticism of $\Phi$-DVAE is that the specification of the model is  restrictive in comparison to relevant approaches, and indeed, we agree this is the case. However, our approach is necessarily restrictive to allow for physically meaningful model-data synthesis, in scenarios where inference on the latent physical system is of interest.

We agree that a model with the true latent dynamics should excel in these examples, which is confirmed experimentally in terms of the MSE and ELBO metrics we report. We recognise this does prevent a truly fair comparison, and the comparisons that we have made are to confirm we do indeed out-perform similar dynamic variational autoencoders (DVAEs). The novelty of the method is the embedding of a particular form of differential equation into the latent space (potentially with unknown dynamic parameters), which is essentially restrictive. The purpose of restricting the latent space in this way is to provide physically interpretable latent inference, with the ability to learn physically meaningful parameters of the model. With KVAE for example, a general linear SSM is assumed, which allows for latent state transitions to be learned, but prevents synthesis with an assumed physical model, or inference of physical parameters.

Again, we thank the reviewers for their feedback and look forward to further discussions.

---

### Decision · Program_Chairs · 2023-01-20

**Decision:**

Reject

**Justification For Why Not Higher Score:**

The positioning of the paper remains ambiguous, which weakens the experimental validation of the contribution.

**Justification For Why Not Lower Score:**

N/A

**Metareview: Summary, Strengths And Weaknesses:**

The paper has received mixed reviews, with three reviewers recommending rejection (5-3-5) and one reviewer recommending acceptance (8). The author-reviewer discussion was fruitful and the authors addressed many of the reviewers' concerns.

The reviewers discussed the unclear positioning of the paper during their private discussion. They noted that if the paper's positioning is in the field of probabilistic sequential modelling, then some of the concerns of 66rK and 91wk remain -- and 8UxT acknowledges those. However, if the positioning is in ML x Physics, then relevant baselines and comparisons should be made to assess the paper's contributions better. Regardless of the actual positioning of the paper, the significance and experimental validation of the contribution thus need to be better demonstrated.

Overall, the reviewers recommend a better positioning of the paper as well as an experimental validation that would be more consistent with it. For these reasons, they believe that the paper is not yet ready for publication.

**Summary Of Ac-Reviewer Meeting:**

The author-reviewer and the private reviewer discussion were sufficient to reach a recommendation.